# Informing disaster-risk management policies for education infrastructure using scenario-based recovery analyses

Eyitayo A. Opabola [1] ✉ & Carmine Galasso [2]

Recent natural-hazard events have shown that post-disaster education continuity is still a significant global challenge. Here, we propose a methodology to support various stakeholders in quantifying the impact of disaster management policies on education continuity in low- and lower-middle-income countries. We then apply the proposed methodology to a hypothetical earthquake scenario impacting a testbed education infrastructure in Central Sulawesi, Indonesia. This case study accounts for local practice influencing recovery through interviews with stakeholders involved in post-disaster management in the region. The analyses reveal that early response financing mechanisms can help speed up education recovery by a factor of three. Also, community-managed school reconstruction projects are likely to be completed up to three to five times faster than agency-managed projects. Furthermore, we demonstrate how the framework can be used to prioritize school reconstruction projects to ensure inclusive education continuity at the community level.

Despite advances in natural-hazard risk understanding, modeling and quantification, and global initiatives to reduce disaster risk to the education sector[1], many countries remain highly exposed to school infrastructure physical damage and severe education disruption from natural hazards. This is especially valid for low- and lower-middle-income countries (as defined by the World Bank[2]). For example, nearly 5000 schools were destroyed following the 2010 moment magnitude 7.0 (M7.0) Haiti earthquake[3]. Over 9000 school buildings and 35,000 classrooms were significantly affected by the 2015 M7.8 Gorkha earthquake in Nepal, with another 7000 schools requiring reconstruction[4]. Natural-hazard-induced physical damage to school buildings, neighboring infrastructure systems, and communities can significantly disrupt education. For example, school closures due to damage and/or inaccessibility affected the education continuity of about 184,000 and one million school pupils in Central Sulawesi, Indonesia (following the 2018 M7.5 earthquake and tsunami) and Nepal (following the 2015 M7.8 Gorkha earthquake), respectively[5,6].

Schools can play a vital role in disaster preparedness, response, and recovery[7–9]. For example, in a pre-disaster setting, school facilities can be used as sites for disaster-preparedness learning activities. In post-disaster scenarios, schools can serve as relief centers, supply, storage, and communication hubs. Hence, community resilience relies on the ability of schools to have efficient disaster preparedness, response, and recovery management strategies. Such recovery management strategies must also ensure post-disaster school continuity.

Several studies have highlighted the importance of post-disaster school continuity. There are various unintended social and economic consequences of education disruption to schools, students, teachers, their families, and the community at large. For example, evidence[6,10] shows that out-of-school children are susceptible to various forms of exploitation (including child labor) and violence (especially in temporary camps), with severe effects on children's long-term development. In addition, it has been reported that education disruption in school children may lead to long-term reduced physical and mental health, leading to a loss of productivity and earnings in adulthood[11,12]. The socioeconomic conditions of staff of closed schools may also be negatively impacted if they need to find alternative jobs (in an already chaotic post-disaster situation) to make ends meet. Also, parents (and

[1]Department of Civil and Environmental Engineering, University of California, Berkeley, USA. [2]Department of Civil, Environmental, and Geomatic Engineering, University College London, London, United Kingdom. ✉e-mail: tayo@berkeley.edu

carers) may have to spend time off work to take care of their children, resulting in a significant productivity loss for the economy[13] as well as well-being losses for them[14]. For nations, the combined influence of education disruption on school children, staff, and their families results in up to a 6% loss in future gross domestic product[15–17]. Hence, post-disaster education continuity must be a priority for any nation.

There are two distinct domains of post-disaster school recovery necessary for education continuity—physical and non-physical. The former is related to the conditions of the physical infrastructure (e.g., classrooms, laboratories, water, sanitation and hygiene (WASH) facilities, power, and water utilities). The non-physical domain, for instance, is associated with the post-disaster management structure and psychosocial recovery of school children and staff[18–20]. Poorly managed disaster-induced psychological disorders can influence changes in behavior, memory, and development of school children[21]; thereby impacting education continuity. There are linkages between the physical and non-physical (especially psychosocial) domains of school recovery. As emphasized by past events, prolonged stay in temporary housing settlements and delayed recovery in school physical infrastructure can impact the long-term psychosocial well-being of school children[19]. We note that our study focuses on the analytical modeling of the physical infrastructure domain of school recovery. Additional studies are needed to explore the efficient integration of physical and non-physical domains of school recovery in analytical recovery modeling frameworks.

The 2015–2030 *Sendai Framework for Disaster Risk Reduction*[22] calls for enhanced disaster risk governance for effective response and the need to "Build Back Better" in sustainable recovery, reconstruction, and rehabilitation. Furthermore, Goal #4 of the 2030 Agenda for Sustainable Development advocates for inclusive education continuity for all[23]. Due to the high physical vulnerability of school infrastructure and the social vulnerability of school children, it is crucial to ensure that sensible policies enhancing post-disaster education continuity are in place. However, various studies have highlighted that governments do not generally prioritize post-disaster education continuity, leading to severe education disruption or even termination. For example, many school children dropped out due to unavailable school infrastructure following the 2018 Central Sulawesi earthquake[24].

In comparison with recovery modeling studies on other infrastructure systems (e.g., residential and business building stock[25], hospitals[26], utility networks[27]), fewer research studies have been carried out on post-disaster recovery of physical education infrastructure. We also note that post-disaster recovery modeling frameworks cannot be generic because building functionality strongly depends on the specific occupancy type. For example, a moderately damaged residential building may be suitable as a shelter-in-place (meaning occupants are not entirely displaced). However, a similarly damaged building might be unsuitable for learning purposes.

On the qualitative side, various resilience-enhancing policies for school infrastructures have been proposed by different studies in recent years[28,29]. Quantitative studies on the resilience of physical education infrastructure started gaining widespread attention in recent years. Available quantitative studies are either field-data-based or simulation-based. Field-based studies[30–33] have emphasized the prolonged post-disaster school reconstruction process and its negative influence on education continuity. The former has been attributed to funding delays, contract issues, the use of schools as temporary shelters by local communities, political setting, land acquisition issues (in cases where school relocation is needed), inaccessible roads for transporting materials to remote locations, management type (i.e., community-managed or agency-managed projects), lack of skilled labor, and flawed planning processes.

Studies have proposed simulation-based probabilistic frameworks to simulate the post-disaster recovery of education systems. Some of these studies[34,35] have developed probabilistic frameworks to simulate the post-disaster recovery trajectory of school infrastructure. However, these studies do not consider the influence of the previously mentioned sociocultural, technical, economic, environmental, and political factors that significantly influence the recovery trajectory of schools in marginalized communities, particularly in low- and lower-middle-income countries. This may be attributed to the fact that such studies are primarily designed to target developed communities (e.g., the USA). Hence, the applicability of such frameworks to tackle a wide range of multi-dimensional issues within a marginalized community may be limited. Herein, we define marginalized communities as groups that experience discrimination and exclusion, especially in pre-disaster mitigation and post-disaster recovery management policies, because of unequal power relationships across social, economic, political, and cultural dimensions.

Fewer studies have sought to contribute to post-disaster recovery modeling in lower-middle-income countries. For example, Alisjahbana et al.[36] developed an optimization approach for school reconstruction scheduling by minimizing the sum of the distance all students in the region have to travel until all schools in the region are reconstructed. However, it is noted that this approach's applicability is limited when critical issues such as land availability, construction site accessibility, and school level can influence reconstruction projects. Hence, a multicriteria decision-making (MCDM) approach that accounts for the influence of various factors on school reconstruction prioritization is needed.

We aim to contribute to the field of critical infrastructure resilience by (1) proposing a post-disaster recovery modeling approach for education systems; (2) demonstrating how the proposed approach can support stakeholders in quantifying the impact of policies (such as early response financing mechanisms and recovery management type) on education continuity. Our proposed framework explicitly incorporates an approach to account for sociocultural, technical, economic, environmental, and political factors influencing the sustainable recovery of school physical infrastructure in low and lower-middle-income countries. First, the recovery time estimation module accounts for various recovery enhancing and impeding factors through a stochastic Program Evaluation and Review Technique (PERT), which enables users to simulate desired levels of pessimism/optimism on each task in the recovery process. Furthermore, the proposed approach embeds a novel MCDM module for intervention prioritization, which accounts for factors such as available intervention budget, land availability, reliance on temporary learning centers (TLCs), age group of students, enabling decision-makers to better account for many factors/criteria that impact recovery in marginalized communities and for their preferences towards those criteria. We first summarize the modeling approach. Then, we adopt a case-study application to demonstrate how end users can use the proposed framework to identify policies for enhanced educational resilience to disasters. For this purpose, a testbed school infrastructure system is developed from a database of schools in Central Sulawesi, Indonesia[37], and is subjected to a hypothetical M7.0 earthquake event. The case study in this paper benefits from insights into the recovery process following the 2018 Central Sulawesi earthquake from engagement with multiple stakeholders under the auspices of the UK Research and Innovation (UKRI)-funded 'Resilient School Hubs' project. The project was approved by the University College London research ethics committee (UCL Ethics Project ID Number: ID280898). The case-study application clearly shows the benefit of early response financing mechanisms on education recovery. Furthermore, the case study highlights the need for NGOs to rethink their approach to collaboration with host communities during the post-disaster recovery process.

## Results
### Post-disaster recovery modeling framework
The proposed methodology combines five distinct modules to evaluate the probabilistic post-disaster recovery trajectory of school

infrastructure. The interdependence of education and utility lifelines (e.g., water and power networks) is not discussed here because (1) past events have shown that utility networks are quickly fixed following disasters[38]; (2) due to climate conditions and architecture (i.e., large windows), most schools in tropical countries are not significantly dependent on the power supply; and (3) some of these schools in tropical countries rely mainly on local wells, not municipal water networks[39]. Interested readers are referred to other studies[40] for more information on simulating the interdependence of buildings and utility lifelines.

The first module of the proposed framework entails a hazard analysis to simulate the local hazard intensity measures (e.g., earthquake-induced ground shaking, flood-induced water depth, typhoon-induced wind speeds) at each school-building location for a particular hazard scenario (i.e., a given event). The hazard analysis adequately accounts for the spatial distribution of the intensities throughout the region of interest. The second module of the proposed framework is a post-disaster functionality assessment, which entails simulating the damage state of each school building conditioned on the site-specific hazard intensity measure. Then, each school's resulting post-disaster functionality level can be estimated depending on the damaged state of each building. The primary functionality indicator considered in this study is education continuity. According to the Comprehensive School Safety (CSS) Targets and Indicators developed by the Global Alliance for Disaster Risk Reduction and Resilience in the Education Sector (GADRRRES)[41], the main post-disaster education continuity indicators include: (1) duration of disaster-induced school closure; (2) the number of students displaced from schools; and (3) number of students in TLCs. In line with the CSS indicators, we define functionality level as the proportion of students with continued access to education (either in a permanent or temporary learning center). For each school, this is estimated as the ratio of students with safe and occupiable classrooms to the total number of students in the school. The third module is a decision-making analysis used to assess the feasibility of education continuity in schools using rapid response strategies such as class scheduling, construction/availability of TLCs, and transfer of displaced students to neighboring schools. These rapid mechanisms' feasibility depends on government policies, available finance mechanisms, and other socioeconomic and political factors. The fourth module accounts for the fact that, in many cases, decision-makers want to achieve competing goals under rigid constraints on time, budget, and workforce to decide on school intervention prioritization. Also, other factors, such as the availability and residual lifespan of temporary school structures, the proximity of displaced pupils to neighboring schools with full functionality, and the number of displaced pupils, are essential. Hence, the fourth module is a MCDM analysis (accounting for the factors above) for intervention prioritization in schools with reduced functionality levels. Lastly, the fifth module is a recovery model used for simulating the recovery time of each damaged school building, accounting for the influence of sociocultural, technical, economic, environmental, and political conditions on post-disaster recovery.

The probabilistic post-disaster recovery analysis generates realizations of recovery trajectories ($Q(t)$), quantifying how quickly the functionality is restored (rapidity) from the initial post-disaster functionality level $Q_o$, and the functionality recovery time $t_R$ (Fig. 1). The model details are provided in the "Methods".

## Post-disaster functionality in the testbed community

Although the proposed framework is general and can be applied to any natural hazard of interest, we specifically consider the post-disaster recovery of a community of 80 schools subjected to a significant earthquake event (Fig. 2). The attributes of the schools (including population, location, and building characteristics) are heavily based on the extensive database of 2500 schools collected by the authors in

Central Sulawesi[42]. The selected 80 schools represent the number of schools within two districts. The decision to select a relatively small testbed is based on the concept that disaster management decisions are generally made at the local government level[43]. Furthermore, the small testbed enables the visualization of the impact of each considered policy at the building level rather than considering low-resolution information over a large grid area. We note that the Central Sulawesi region is prone to cascading hazards. Given the focus of the case study on earthquake-induced ground shaking, the main testbed selection criteria are the low potential for earthquake-induced cascading hazards (e.g., liquefaction, landslides, and tsunami) and the reliability of available information on the schools. An inter-rater analysis was used to assess the reliability of the available information[42].

A total of 280 school buildings are in the selected 80 schools. The 280 school buildings serve 17,055 pupils from the considered community. Of the 80 schools, 51 are primary, and 29 are secondary (Fig. 2). In addition, 89% and 11% are one- and two-story buildings, respectively. The one-story and two-story buildings are confined masonry and infilled reinforced concrete frame buildings, respectively. In addition, 75% and 25% of buildings are assumed to be designed/built pre- and post- the updated Indonesian seismic code (SNI 1726:2012)[44], respectively. The number of buildings in the 80 schools ranges from one to nine, with a median of three and a standard deviation of 1.4. The average pupil population in each school building is 61, with a standard deviation of 16.5. For this study, we assume that all schools are state-owned and the same policies apply to all. We note that the boundary around the 80 schools in Fig. 2 is hypothetical (to guarantee the anonymity of each specific school).

The case study is carried out for an M7.0 earthquake scenario assumed to occur on a hypothetical North-Northwest-South-Southeast (NNW-SSE) trending strike-slip fault (Fig. 2). A $V_{s30}$ of 300 m/s is assumed for the entire community. One thousand realizations of spatially cross-correlated intensity measures (IMs) (i.e., spectral accelerations) at the building locations (and for the building fundamental periods of vibration) are generated, as described in the "Hazard Analysis" section of "Methods". The post-earthquake functionality assessment follows the modeling approach presented above and described in detail in the "Methods". Building-level fragility models are used for the post-earthquake functionality assessment. Due to the absence of fragility models for schools in Palu, we selected fragility models based on a review of published studies from similar archetypes in South Asia and globally[45–47]. Based on expert judgment, we concluded that the fragility models from these countries could be adopted in Palu. In addition, we note that the study aims to showcase the proposed framework using realistic input data and discuss the relative effect of various disaster-risk management policies rather than perform a detailed/realistic risk assessment. Additional studies are, however, recommended to develop fragility models for school buildings in Central Sulawesi.

## Development of case-study scenarios

We now demonstrate how the proposed framework can support stakeholders with disaster management planning by quantifying the influence of various policies on the disaster recovery of their education infrastructure. To ensure we address realistic problems, the case-study scenarios were highlighted from focus group discussions and interviews with school principals, NGO officials, government officials, engineers, and contractors involved in the 2018 Central Sulawesi earthquake recovery projects[48]. Information related to the stakeholder engagement exercises (including details of the area of inquiry, stakeholder engagement type, guiding questions, and the number of stakeholders) is available online[48].

First, school stakeholders noted that insufficient anticipatory funding was a critical factor in the lack of classrooms for school pupils after the 2018 event. Furthermore, several schools had to rely on local

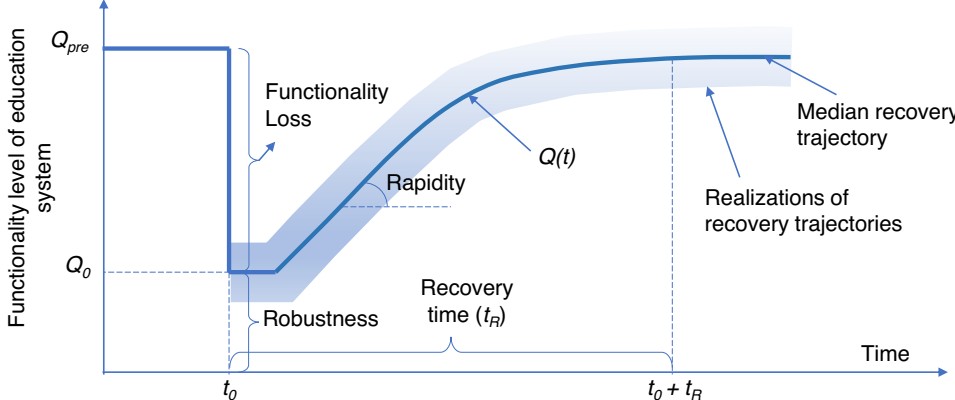

**Fig. 1 | Graphical representation of a community-level recovery trajectory.** $Q_{pre}$ is the pre-disaster functionality level; $Q_o$ is the initial post-disaster functionality level immediately after the event (i.e., at time $t_O$). The recovery time $t_R$ is the time to restore full functionality to the system. Given that the analysis is probabilistic, the generated recovery trajectory is also probabilistic in nature (i.e., multiple realizations of the recovery trajectory are simulated).

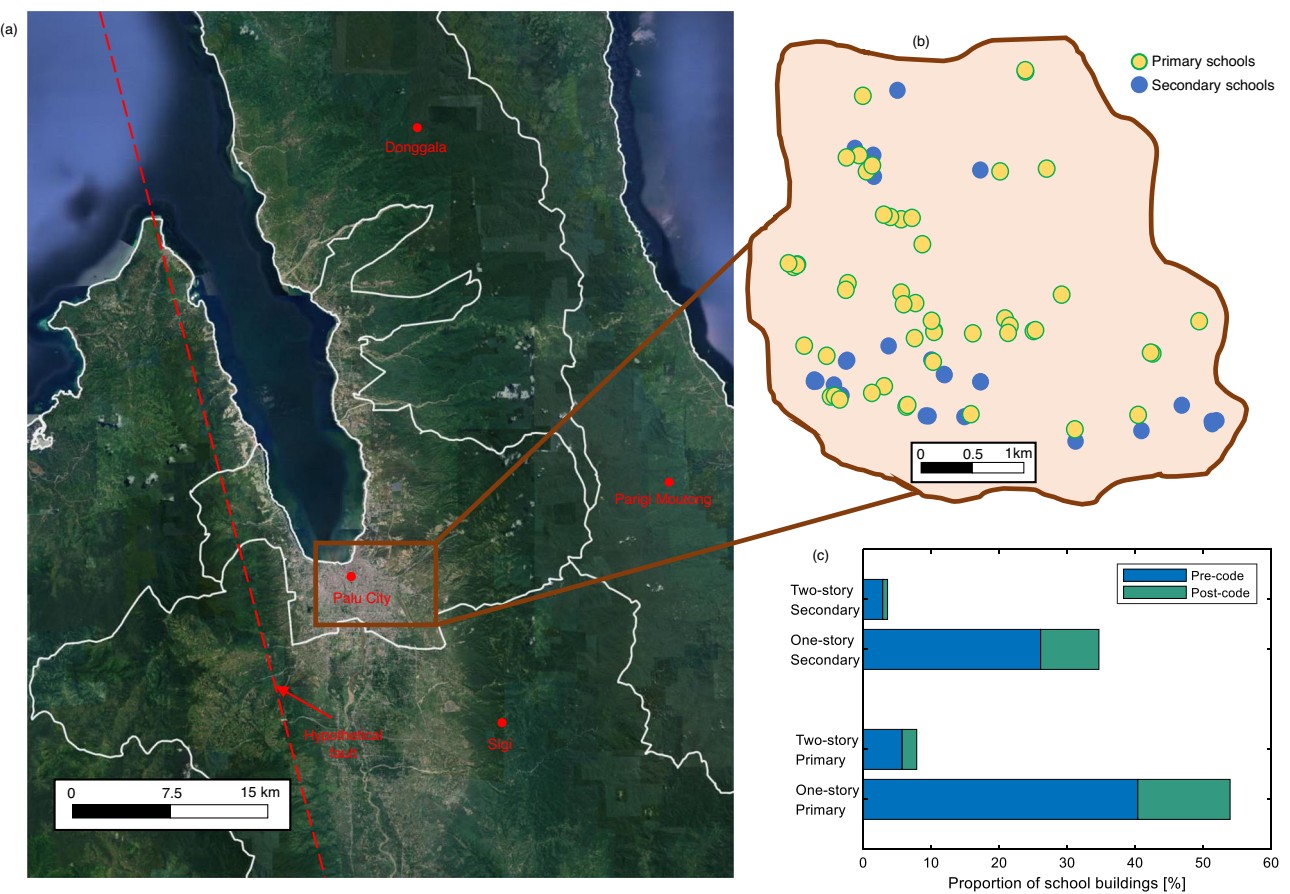

**Fig. 2 | Testbed school community adopted in the study. a** The map of Central Sulawesi showing Palu City, Sigi, Donggala, and Parigi Mountong regencies. The hypothetical fault is shown with a dashed line. **b** The relative location of all 80 schools in the testbed. **c** The distribution of the number of stories, age level, and design era of school buildings is shown at the bottom left. Pre-code and post-code buildings were constructed before and after the SNI 1726:2012[44] building code. Credit: Imagery ©2023 TerraMetrics, Map data: Google ©2023.

and foreign aid to build TLCs. Stakeholders noted that school communities might respond better to disasters if there were tools to forecast the amount of anticipatory funding required.

Another issue highlighted was the delay in reconstructing permanent school buildings in projects handled by NGOs. Due to this delay, almost four years since the earthquake, several schools still use TLCs (typically with a lifespan of 4–5 years). As a result, several schools

may be susceptible to increased education discontinuity if TLCs start experiencing damage and loss of functionality.

Lastly, NGO officials discussed that they do not have a systematic method of selecting schools to focus on for their reconstruction projects. The primary criterion considered was the availability of well-cleared land for their projects. Due to delays in NGO projects, this approach may be unfavorable to schools with significant reliance on

TLCs, especially if such schools could have otherwise benefitted from other management types (e.g., community-led management).

The subsequent subsections in this paper adopt the three highlighted discussions in demonstrating the applicability of our proposed framework to capture multi-dimensional issues affecting recovery.

## Influence of available early response financing mechanisms on education continuity

The most appropriate way to achieve a desirable post-disaster education continuity level is through the effective retrofit/reconstruction of school buildings in a pre-disaster scenario. However, several school buildings will remain vulnerable due to a lack of financial resources and sufficient technical know-how to identify disaster-prone structures. A helpful way for local authorities to plan for post-disaster education continuity is through early response financing mechanisms[49]. Such funding can be used for interventions such as repairing and retrofitting damaged school buildings, constructing TLCs, and buying relevant learning materials.

Using the proposed framework, we demonstrate how local authorities can estimate a given school's education continuity recovery trajectory (through the construction of TLCs), accounting for the uncertainties involved in the process. We assumed that because they are 'makeshift' structures (i.e., tents), TLCs are typically non-engineered and do not require skilled labor or heavy equipment for their construction. Hence, the technical factors that affect the construction of permanent structures (where building permits are needed, the tender process may be necessary, and lack of building materials and technical know-how are more influential) are not significant for TLCs. Although not considered in this analysis, we note that the proposed methodology can capture the influence of other recovery-impeding factors on the construction of TLCs.

Engagement with the school principals[48] showed that some schools could carry out immediate post-disaster intervention using anticipatory funding set aside from the school operational assistance funding (locally referred to as Bantuan Operasional Sekolah (BOS)) provided by the government. More information on BOS can be found online[50]. We observed that not all schools had this anticipatory funding, which impacted their recovery.

Figure 3 illustrates two cases of available anticipatory funding for the schools subjected to the considered M7.0 scenario for the median IM realization. The first is a case where 45% of the 80 schools have anticipatory funding. Without any model to simulate the capability of principals to set aside anticipatory funding, we randomly assigned anticipatory funding to 36 schools in the testbed. In reality, such data

can be collected by surveying school principals in a region of interest. Based on field data from the 2018 Palu earthquake, we assume that the average time to construct a TLC with early finance mechanisms is 40 days after the disaster. Using a time mitigation factor of 0.5 (i.e., for community participaton - described in "Methods"), the optimistic time $a$ is taken as 20 days. The pessimistic time $b$ is taken as 80 days by assuming a time amplification factor, associated with delay in material procurement, of 2.0. The most likely time $m$ is taken as the average of the sum of $a$ and $b$ (see discussions in "Methods"). For schools without access to early response financing mechanisms, we assume that $a$, $m$, and $b$ are further amplified by a factor of 2.0 (i.e., due to delays in funds disbursement) – i.e., $a = 40$ days, $m = 100$ days, and $b = 160$ days. As shown in Fig. 3a, in a case where only 45% of the 80 schools have early response financing mechanisms, only a small fraction of the schools can ensure education continuity within two months, and the remaining schools may need to rely on foreign aid to construct TLCs, resulting in an undesirable recovery rapidity level. On the other hand, sufficient anticipatory funding (i.e., all 80 schools have anticipatory funding) allows all the damaged school buildings to be immediately replaced by TLCs. For the case-study community, we show that sufficient anticipatory funding can reduce the recovery time for education continuity (through rapid construction of TLCs) by a factor of up to three.

One of the advantages of this scenario analysis is that local authorities can easily use a target hazard scenario, given the vulnerabilities of schools in their communities, to plan for sufficient early response financing for TLC construction while still making pre-disaster efforts to mitigate the vulnerabilities of the schools (e.g., retrofitting).

## Influence of management type on the reconstruction time for permanent school buildings

The government, NGOs (both local and international), or the host communities typically provide/contribute to post-disaster school reconstruction project funding. Post-disaster recovery reports[33] in major global disasters show that most permanent school-building reconstruction projects are either community- or agency-managed. The community-managed approach is a locally implemented approach where the construction management (e.g., material acquisition, selection of building contractors) is led by local authorities, school management committees, or host community leaders. The agency-managed approach is a case where a government or non-government entity leads the construction management.

Several field-based comparative studies[33,48,51–54] have highlighted the successes and weaknesses of community-managed and agency-managed reconstruction projects (residential and school buildings) in

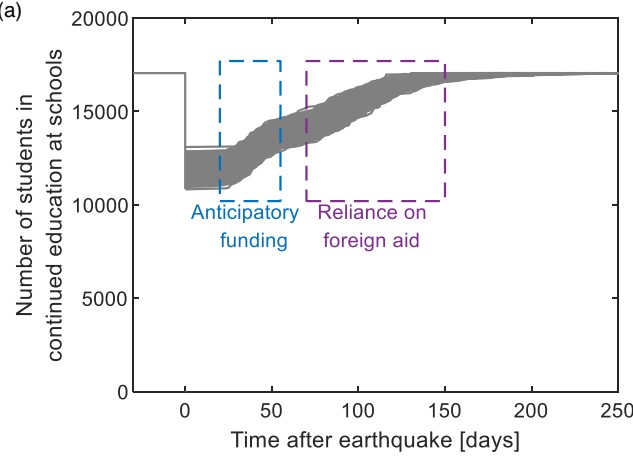

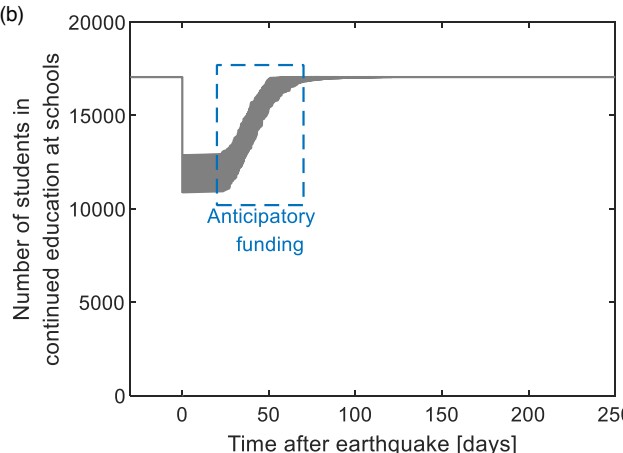

**Fig. 3 | Recovery trajectories for the considered budget availability scenarios for the median ground-motion intensity field. a** The effect of minimal anticipatory funding (i.e., only 45% of all schools have funding); **b** the effect of

sufficient anticipatory funding (i.e., all schools have funding). The dashed boxes show the influence range of the funding mechanisms.

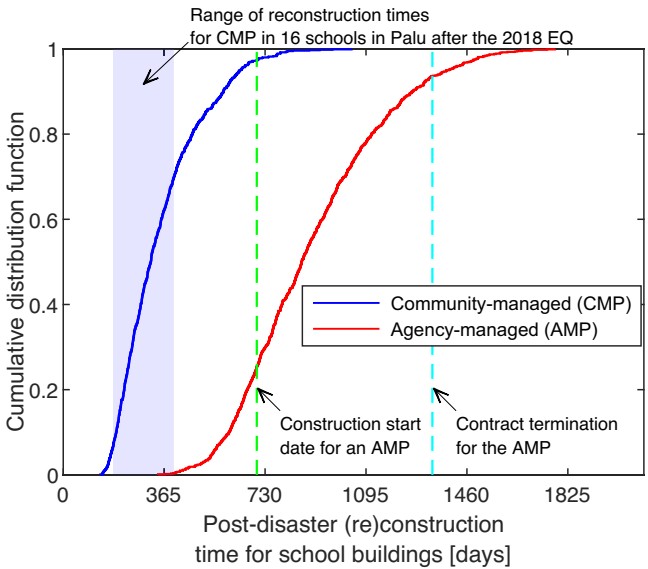

**Fig. 4 | Impact of management type on reconstruction time of permanent school buildings.** Simulated cumulative distribution function (CDF) of the reconstruction time for community-managed (median of 310 days) and agency-managed (median of 860 days) projects. The reconstruction times in the testbed community are estimated using the recovery time model presented in Eqs. (3)–(5). The CDF accounts for recovery-impeding (i.e., time amplification) factors that may influence both community- and agency-managed projects. The community-managed school building projects have a lower median recovery time because of little to no internal bureaucratic issues, more coordinated arrangements with relevant stakeholders, and little to no delay in a contract bidding process in these types of projects. The range of completion times for community-managed building reconstruction in 16 schools in Palu (after the 2018 earthquake) is shown in the blue box. The green and cyan lines show the start and contract termination date for one of the prominent NGOs handling reconstruction projects in over 20 schools in Palu[48].

lower-middle-income countries. For example, the abundance of local knowledge and experience positively influences the recovery process of community-managed projects and achieves higher beneficiary satisfaction. Also, the fact that the drivers of community-managed projects (i.e., local authorities, school management committees, or host community leaders) typically reside in the communities is a strong motivation to ensure higher quality and rapidity in reconstruction projects. On the other hand, the agency-managed approach has been described as a 'one-size-fits-all' approach intended to suit donors and implementing agencies and rarely involves the target beneficiaries[51,52]. Typically, agency-managed projects are subjected to delays resulting from internal bureaucracy, agreements with the local authorities, community, and school authorities, and the lengthy bidding process for engineers and contractors. Nevertheless, regardless of the management type, a given time is spent on damage assessment, clearing the site of rubbles (or identifying a relocation site), getting the relevant building permits, and so on. We note that principal interviews and focus group discussions with key stakeholders (engineers, contractors, government, and NGO officials)[48] actively involved in the recovery process of the post-2018 Palu earthquake highlighted significant delays of agency-managed projects in Palu.

Considering all these impeding factors in the recovery modeling framework, the cumulative distribution function (CDF) of the simulated post-disaster reconstruction time for a school building from community- and agency-managed projects can be assessed as presented in Fig. 4. We explicitly account for the influence of delayed bidding process, internal bureaucracy on the bidding, and construction mobilization of agency-managed projects through recovery time amplification factors described in "Methods". In line with the earlier

discussions, these challenges are assumed to be less prominent in community-managed projects and are not considered in the recovery time modeling process (See Table 2). In a pre-disaster scenario, the required time to construct a typical one-story school building in Palu is 4–6 months[48] (see "Methods" for more information). As shown in Fig. 4, community-managed projects, on average, would be completed more than two times slower than in a pre-disaster scenario.

We note that we interviewed 16 school principals with reconstruction projects in their schools and discovered that the completion time ranged from 180 to 400 days[48]. This range is effectively captured in Fig. 4, providing some evidence of the validity of our proposed framework. We can compare the observed recovery time of collapsed buildings in Palu with the simulated recovery time because the recovery time models are conditioned on the post-earthquake damage state and not the earthquake scenario (i.e., ground-motion intensities).

On the other hand, as shown in Fig. 4, agency-managed projects are likely to be completed 2–4 years after the disaster. We also assessed ongoing agency-managed reconstruction projects in Palu by interviewing NGO officials and contractors. For an unnamed NGO with a significant number of school reconstruction projects in the Palu region, most reconstruction sites have not yet been reached four years after the disaster. More details are provided in Opabola et al.[48]. Hence, constructing permanent structures in those schools may take up to five years or more.

A key recommendation from this case study is that the government needs to understand that agency-managed projects may be delayed. As such, local authorities may only consider allowing NGOs to manage the reconstruction of schools that a prolonged reconstruction process would impact less. Further discussions on this are provided in the subsequent case study scenario. Furthermore, the efficiency of agency-managed projects can be improved through more trust and collaboration with local communities. For example, NGOs may choose to disburse funds to support efficient community-managed reconstruction projects rather than manage projects themselves.

## Accounting for the influence of sociocultural, environmental, and political on the recovery process

Apart from the typical technical and financial delays in the recovery process, sociocultural, environmental, and political issues may impede or speed up recovery. For example, following the 2004 Indian Ocean tsunami, recovery projects in conflict zones in Sri Lanka were eight times slower than in areas without conflicts[55]. Being able to account for such sociocultural and political factors can enable local authorities to understand how existing issues could further impede recovery in their zones. The proposed methodology accounts for these conditions through amplification factors calibrated using data from past events. These factors are applied to the relevant intervention process, as described in the "Methods" section.

Using the proposed methodology, we explore how a combination of sociocultural, environmental, and political issues can influence the average reconstruction time for community-managed and agency-managed projects. Five factors were considered—(1) hostile political conditions; (2) pandemic; (3) land disputes (for relocation projects); (4) poor management skills of contractors; (5) a combination of land disputes and poor management skills of contractors. For this analysis, the average mobilization and inspection time for the community- and agency-managed projects are combined with time amplification factors using Eqs. (3)–(5) in a probabilistic manner (see description in "Methods").

Figure 5 presents the output of the probabilistic analysis and shows how various sociocultural, environmental, and political conditions can impede school reconstruction projects. For the considered scenarios, hostile political conditions resulted in the longest recovery time for permanent school buildings. As shown in Fig 5, the estimated

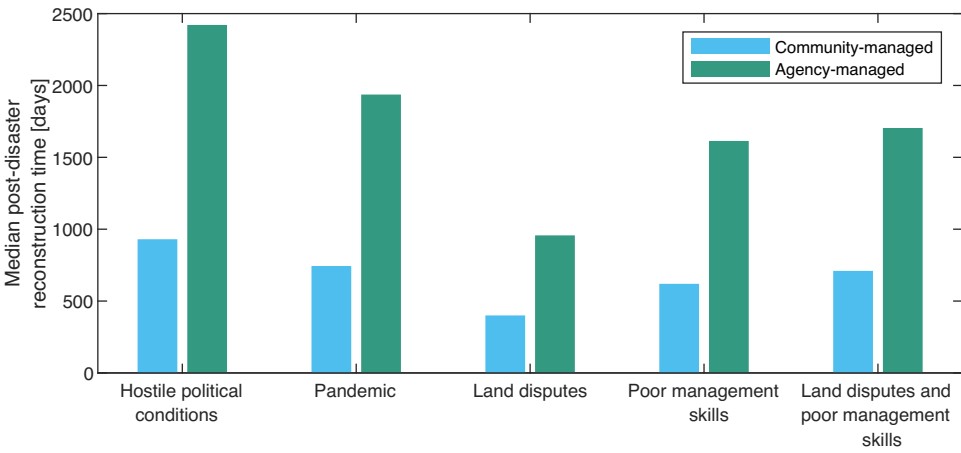

**Fig. 5 | Influence of sociocultural, environmental, and political factors on the median post-disaster reconstruction time for school buildings.** The following amplification factors ($\beta$) were used in the simulation: hostile political conditions ($\beta = 5.0$); pandemic ($\beta = 3.0$); land dispute ($\beta = 2.0$); poor management skills of contractors ($\beta = 3.0$); a combination of land dispute and poor management skills of contractors ($\beta = 3.0$). The time amplification factors used in simulating the influence of these conditions in Eqs. (3)–(5) are based on Supplementary Table 3. For each of the five scenarios, the optimistic time ($a$) was estimated using a time mitigation factor of 0.5, while the pessimistic time ($b$) is based on the upper-bound values of the time amplification factors in Supplementary Table 3, and $m$ (most likely time) is taken as $0.5(a+b)$.

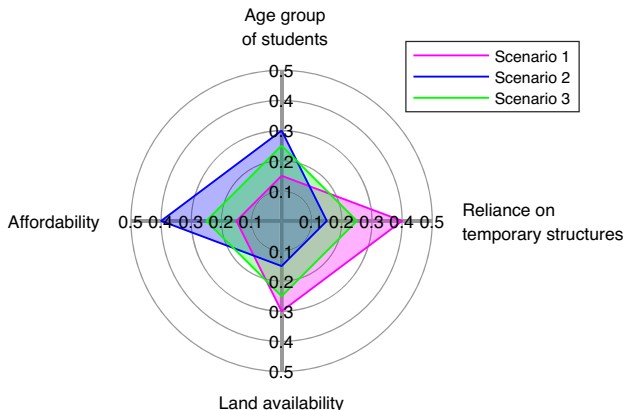

**Fig. 6 | Criteria weights for the considered scenarios for the multicriteria decision-making analysis.** Scenario 1 is a case where decision-makers consider the level of a school's reliance on temporary learning centers as the primary criterion for identifying schools that need to be prioritized for permanent structures' (re) construction. Scenario 2 is a case where affordability is considered the most important criterion, while Scenario 3 looks at a case where all four criteria are equally weighted.

median recovery time for reconstructing a damaged school building is significantly influenced by the management type and the considered recovery impeding factors. For the considered scenarios, hostile political conditions resulted in the longest recovery time for permanent school buildings. For example, local authorities may decide to encourage only community-managed projects in regions with hostile political conditions based on the information presented in Fig. 5.

### Influence of decision-makers' preferences on school reconstruction prioritization

As mentioned earlier, during the interviews, NGO officials involved in reconstruction projects discussed the lack of a systematic methodology for developing a school reconstruction prioritization list. The proposed framework includes a MCDM module that can support the government or NGOs in creating such a school reconstruction prioritization list. However, it is noted that the prioritization list depends on the weights decision-makers append to each criterion. Therefore, criteria weights for developing the prioritization list are selected based on subjective judgment. Nevertheless, decision-makers can use our methodology to perform sensitivity analyses and see how the prioritization hierarchy fluctuates for a given suite of criteria weights.

Figure 6 describes a case study where decision-makers are interested in four criteria (described in "Methods") and three scenarios. The first scenario is where the decision-makers consider the level of a school's reliance on TLCs as the primary criterion for identifying schools that need to be prioritized for permanent structures' (re)construction. The second scenario examines a case where affordability is considered the most important criterion, while the third scenario looks at a case where all four criteria are equally weighted (i.e., zero bias).

Figure 7a looks at defined performance metrics for the 15 damaged schools with average post-disaster functionality lower than 0.33. Further discussions on how the criteria and performance metrics are defined are provided in "Methods". Using the proposed approach, Fig. 7b provides the reconstruction prioritization hierarchy for each scenario. As shown in Fig. 7b, the selected weights significantly influence the list. As shown in Fig. 7a, School 15 is an elementary school that relies heavily on temporary structures, but the reconstruction project would take 27% of the available budget. For scenario 1 (preference for schools relying on temporary structures), School 15 gets prioritized (Fig. 7b). However, in scenario 2 where the decision-makers choose not to prioritize the most capital-intensive projects, School 15 drops in the prioritization order. The MCDM can enable decision-makers to visualize whether their bias (reflected in criteria weights) results in inclusive recovery.

Local authorities can use the reconstruction prioritization list to designate school intervention projects to different management types. For example, schools ranking low in the prioritization list may be designated as agency-managed projects. It is also noted that the ranking list can always be updated by repeating the analyses whenever local authorities update the performance metrics and/or criteria weights.

## Discussion

We recognize that one of the best ways to enhance the resilience of education systems is through appropriate retrofit of existing school

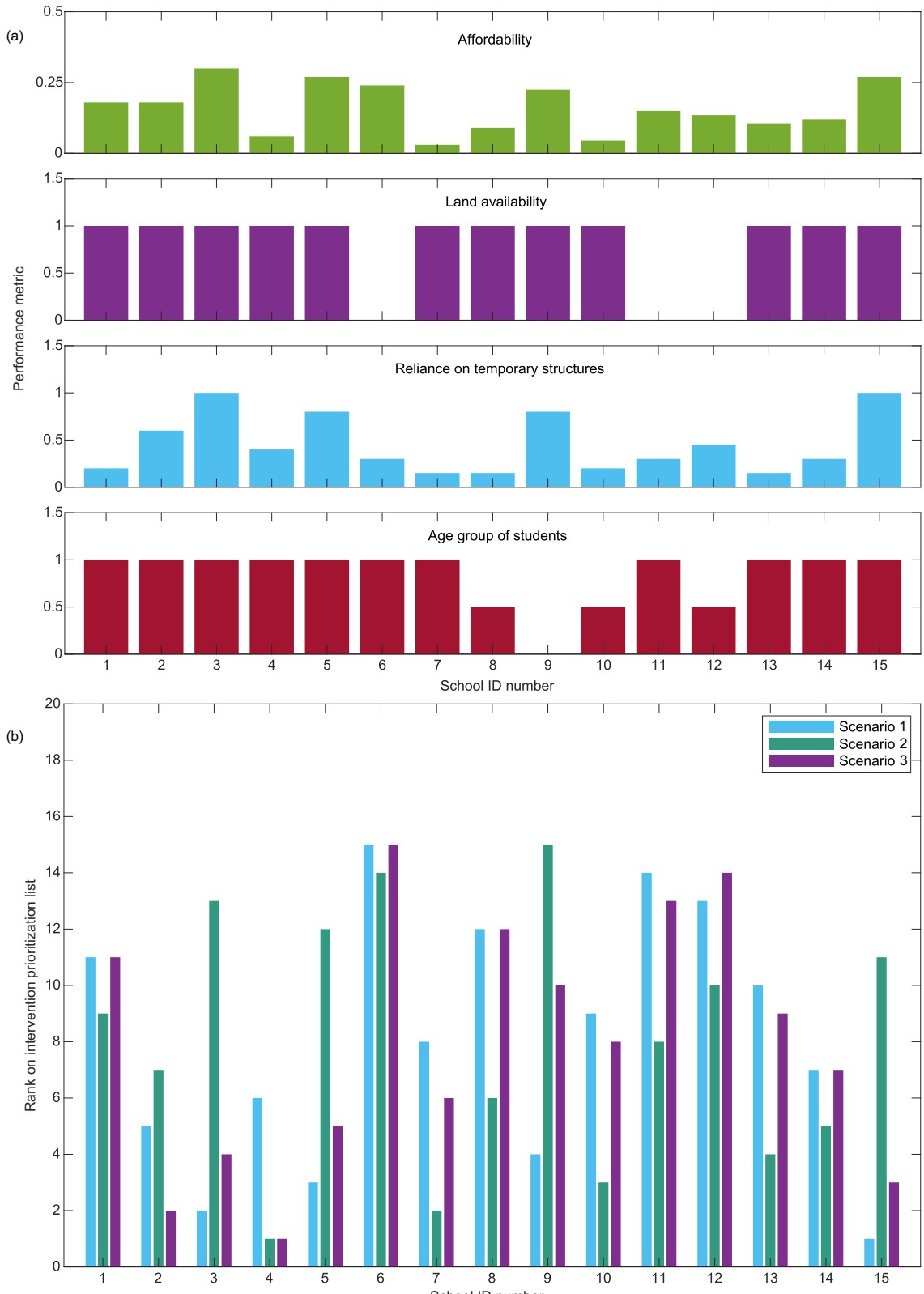

**Fig. 7 | School reconstruction prioritization analysis. a** Performance metrics for the 15 damaged schools used in the multicriteria decision-making analysis. **b** Reconstruction prioritization hierarchy for the 15 damaged schools based on the three considered scenarios. A rank value of one means highest priority and a rank value of 15 means lowest priority. The criteria weights for each scenario are defined in Fig. 6.

buildings before any significant hazard hits. Yet, it is also essential to acknowledge that financial and technical constraints may make reducing the physical vulnerabilities of all existing buildings unrealistic. Moreover, disaster risk reduction is more challenging in low and lower-middle-income countries due to various sociocultural, technical, economic, environmental, and political motivations. Therefore, another approach for enhancing the post-disaster resilience of education systems in low and lower-middle-income countries is through policies

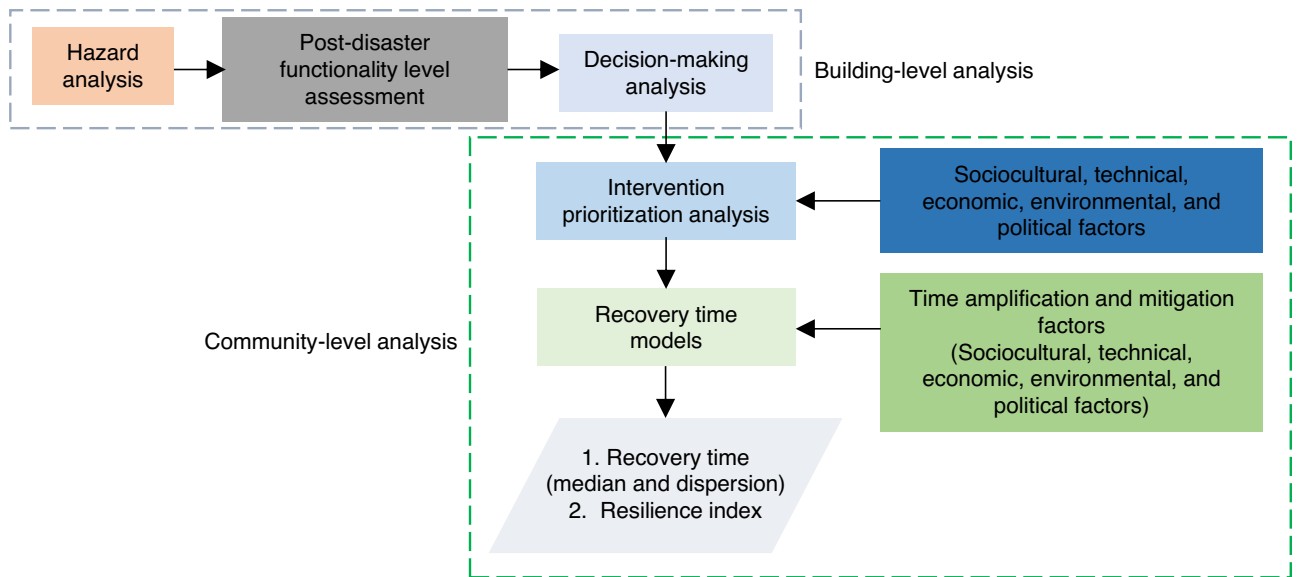

**Fig. 8 | Flowchart for the proposed framework.** The first three steps are building-level modules that define the hazard intensity measure, post-disaster functionality level of impacted buildings, and required post-disaster intervention on damaged buildings. The community-level modules define intervention prioritization and recovery trajectory for damaged schools, accounting for technical, environmental, socioeconomic, political, and cultural factors influencing recovery.

that ensure education continuity, accounting for the vulnerabilities of existing building portfolios.

This study presented a novel framework for simulating the probabilistic recovery trajectory of education systems. The proposed framework accounts for the influence of sociocultural, technical, economic, environmental, and political factors on the post-disaster recovery process of education systems. Local authorities can use the proposed framework to quantify the potential efficacy of several policies, given their limited resources and other local issues in their communities.

We demonstrated the application of the proposed methodology to a hypothetical earthquake scenario impacting a testbed of 80 schools developed from a database of schools in Central Sulawesi, Indonesia. The analyses reveal that early response financing mechanisms can help speed up education recovery by a factor of three. Also, community-managed school reconstruction projects are likely to be completed up to three to five times faster than agency-managed projects.

While the considered case studies focus on financing mechanisms, (re)construction management type, and the influence of socio-political, environmental, and cultural factors on the recovery process, the framework can also be used to simulate the influence of modular construction, retrofit programs, and increased construction workforce on the community-level resilience of education systems. We note that the results presented in this paper reflect observed patterns from the post-2018 Sulawesi earthquake recovery process in schools in Palu, Sigi, and Donggala (Indonesia). Hence, the proposed methodology can support policy-making exercises by adequately considering disaster-vulnerable communities' sociocultural, technical, economic, environmental, and political issues.

## Methods
The proposed methodology and presented results integrate several probabilistic and MCDM analyses to simulate the post-disaster education continuity level and recovery time for education systems at the building- and community levels. The framework flowchart is presented in Fig. 8. The figure shows that the first three steps are building-level analysis modules, while the remaining are community-level analysis modules. The analytical steps are summarized below.

### Hazard analysis
We derived earthquake-induced ground-motion intensity measures (IMs, herein peak ground accelerations (PGA) and spectral accelerations at 0.2 sec ($S_a$ (0.2 sec)) for the one- and two-story buildings, respectively) at each school building site using the Campbell and Bozorgnia ground-motion model[56]. Furthermore, we used the Principle Component Analysis approach[57] to generate 1000 realizations of spatially cross-correlated spectral intensities at the building locations.

### Post-disaster functionality assessment
For each of the thousand realizations of spatially correlated spectral accelerations, the post-disaster functionality level of each building is simulated. In this process, the post-earthquake damage state of each school building is simulated using fragility models quantifying the probability of an asset exceeding a given level of damage (i.e., none, slight, moderate, extensive, and complete) at different hazard intensity values. Buildings with no and slight damage are classified into functionality level zero (FL0) and one (FL1), respectively, and were assumed to be safe for immediate occupancy. Buildings with moderate damage are classified into FL2 and assumed to be uncollapsed but unsafe for occupancy. Lastly, buildings with extensive and complete damage are classified as FL3. The assumed median fragility estimates (PGA) for the pre-code one-story buildings are 0.2g, 0.55g, and 0.60g for FL1, FL2, and FL3, respectively. The assumed median fragility estimates (PGA) for the post-code one-story buildings are 0.50g, 0.90g, and 1.1g for FL1, FL2, and FL3, respectively. The assumed median fragility estimates ($S_a$(0.2 sec)) for the pre-code two-story buildings are 0.50g, 0.90g, and 1.2g for FL1, FL2, and FL3, respectively. The assumed median fragility estimates ($S_a$(0.2 sec)) for the post-code two-story buildings are 1.0g, 1.85g, and 2.30g for FL1, FL2, and FL3, respectively. A lognormal standard deviation of 0.4 was assumed for all fragility models.

The post-disaster functionality of a school $s$ (i.e., the proportion of students with access to a classroom in the school) given a $j^{th}$ simulated ground-motion IM level resulting from an earthquake scenario EQ is then assessed as:

$$q_{0,j}^s \big| IM_j^s, EQ = \frac{n_{st\_FL0,1}^s \big| IM_j^s, EQ}{n_{st\_tot}^s} \tag{1}$$

**Table 1 | Criteria for school intervention prioritization**

|  | Criteria | Category |
|---|---|---|
| $C_1$ | Available intervention budget (Affordability) | Cost |
| $C_2$ | Land availability | Benefit |
| $C_3$ | Reliance on temporary structures | Benefit |
| $C_4$ | Age group of students | Benefit |

where $n^s_{st\_FL0,1}$ is the population of students with access to FL0 and FL1 buildings in school $s$ ($s$ = 1, 2, ..., 80 for the case study), and $n^s_{st\_tot}$ is the total population of students in school $s$.

The community-level functionality (i.e., the proportion of students with access to a classroom in the community) given a simulated ground motion IM level resulting from an earthquake scenario EQ ($Q_{0,j}|EQ$) can also be expressed as:

$$Q_{0,j}|EQ = \sum_{s=1}^{n_s} q^s_{0,j}\Big|IM^s_j,EQ \qquad (2)$$

where $n_s$ is the number of schools in the community.

The community-level functionality can also be expressed in terms of population as the product of $Q_{o,j}$, and the total number of students in the community.

**Decision-making analysis**
Here, we decide on the appropriate intervention phase for school buildings at each functionality level. We assumed that FL0 and FL1 buildings could be occupied during minor repairs to structural and non-structural components. Hence, there is no need for any TLCs. On the other hand, FL3 buildings require the most significant intervention. We assumed that all FL3 would initially be replaced by TLCs, followed by the construction of permanent structures[58]. FL2 buildings are assumed to require temporary closure pending heavy repair and/or strengthening work on damaged components.

**Intervention prioritization analysis**
The intervention prioritization analysis is needed for developing an intervention prioritization list for damaged schools at the community level. However, in cases where each school makes its decision, this analysis is unnecessary. For example, we did not use the intervention prioritization analysis for the case study on the influence of available finance mechanisms at each school for the construction of TLCs.

The intervention prioritization list is developed in this study using the technique for order of preference by similarity to ideal solution (TOPSIS)[59]—a MCDM method. MCDM has been adopted in this study because of its popularity and simplicity. It can be easily implemented by relevant decision-makers (e.g., local disaster risk management authorities, education ministries, and international NGOs)[60]. For the sake of brevity, the calculation steps are not shown here. The inputs for the TOPSIS method are the performance metrics and criteria weights. Four criteria are considered based on stakeholder engagement presented in a separate study[48] (See Table 1). Aside from the four considered criteria, decision-makers may consider others (e.g., proximity to neighboring schools with available space for new students, school management type—i.e., private- or government-owned). However, we do not consider proximity to neighboring schools in this study because other studies have highlighted the negative impact of school mobility (e.g., lower academic achievement, reduced social interactions, and health and developmental problems) on school children[61–63].

The first criterion is intervention affordability. This criterion accounts for the fact that decision-makers may be interested in ensuring the available financial resources are spread over a wide range of schools as much as possible. Therefore, the performance metric is defined as the proportion of the reconstruction cost for each school

relative to the total available financial budget. In the considered case study, we look at a scenario where the reconstruction cost for the entire school community exceeds the available budget (i.e. sum of all the affordability performance metrics is greater than 1.0).

The second criterion is land availability which accounts for the fact that construction work can only occur if the required landmass is available. For example, prioritizing a reconstruction/relocation project for a school with ongoing local land disputes may not be optimal. A binary metric is adopted here. Values of 0 and 1 are adopted for cases without and with land availability, respectively. For the case study, we randomly assumed that three schools are located where post-disaster school reconstruction projects are prohibited, and the government has yet to sort out relocation logistics (i.e., land acquisition) for these three schools.

The third criterion is the availability of temporary structures. The urgency of permanent reconstruction to replace damaged school structures is related to the availability and lifespan of temporary structures constructed after the disaster. For example, quick (re)construction of buildings for schools without temporary structures is essential to ensure education continuity. Also, in the case of schools with temporary structures, it may be necessary to ensure that newly constructed permanent facilities are available by the end of the lifespan of the temporary structures. Supplementary Table 1 presents performance metrics for 'reliance on temporary structures' as a function of the proportion of temporary structures (i.e., percentage of school buildings in the considered school that are temporary structures) and the estimated age of temporary structures at the expected completion of permanent structures. Supplementary Table 1 has been developed with the assumption that the average lifespan of the temporary structures is 4–5 years. In the case study, the estimated age of TLCs at the expected completion date of permanent structures is taken as the median reconstruction time of a permanent school building (i.e., 310 days – see CDF in Fig. 4). It is noted that the estimated median reconstruction time is associated with the considered recovery-impeding factors in the case study. For schools requiring more than one new permanent building, we assume that the age of TLCs at the expected completion date of permanent structures is greater than four years (i.e., based on the assumption that new buildings in each school are reconstructed in sequence). For example, for school ID 2, the performance metric for reliance on temporary structures is 0.6 because 50% of the current school buildings are TLCs.

The fourth criterion is the age group of students in the school requiring intervention. The decision-makers may desire to assign priority levels to different age groups. In this study, we assumed that elementary schools have the highest priority, junior high schools have medium priority, and senior high schools have the lowest priority (Supplementary Table 2).

As in Table 1, we treat three of the criteria as benefit criteria (i.e., an increase in the performance metric would result in a school gaining a higher priority on the intervention list). For example, a school with only (100%) temporary structures on the premises with an average expected age at the end of the reconstruction process greater than four years has a metric of unity (See Supplementary Table 1). This means that such a school would be prioritized over a school in which less than 20% of the school buildings are temporary structures. The intervention affordability is treated as a cost criterion (i.e., an increase in the corresponding performance metric would result in a school having a lower priority on the intervention list).

**Recovery time modeling**
Recovery models are used to simulate the recovery trajectory of a given school building for each realization of ground-motion IMs. The influence of sociocultural, technical, economic, environmental, and political (STEEP) conditions/factors is incorporated in the recovery models using recovery time mitigation ($\alpha$) and amplification factors ($\beta$)

**Table 2 | Average time for various stages of the recovery process for one-story school buildings in Central Sulawesi**

| Recovery phase | Average time $T$ [days] |
|---|---|
| Agreement between agencies and local authorities* | 90 |
| Tender process* | 180 |
| Structural design | 60 |
| Building construction permit | 90 |
| Reconstruction time (includes substructure and superstructure) | 180 |

*Only considered for the agency-managed projects. These values are from semi-structured interviews with NGOs, engineering firms, and contractors actively involved in reconstruction projects in the Palu region

through stochastic network analysis[40]. The average recovery time is assumed to be the sum of the average time required to inspect damaged buildings ($T_{insp}$), for the bidding and construction mobilization ($T_{mob}$), and to restore functionality through selected intervention process ($T_{int}$)—the three considered phases. The optimistic time (i.e., the minimum time required to complete each recovery phase) for each of the three phases for a building $z$ is:

$$a_{i,z}(t)|STEEP,FL_z = \prod_{l=1}^{p} \alpha_{i,l}(t)|STEEP \times T_{i,z}|FL_z \quad (3)$$

where $i$ is the recovery phase, i.e., inspection, mobilization, or intervention; $T$ is the average time to complete a process in a pre-disaster scenario (i.e., without any direct or indirect influence of time-impeding factors), $l$ are the considered time mitigation factors ($\alpha$) ($l = 1, 2, ..., p$) influencing phase $i$; and $t$ is the time since the earthquake occurrence. Recovery mitigation and amplification factors may be time-dependent[40]; hence $a_{i,z}$ is time-dependent. STEEP refers to sociocultural, technical, economic, environmental, and political factors that can either impede or speed up recovery, e.g., construction delays, funds availability, land dispute issues, pandemics, conflicts, and community participation level in the recovery process. Equation (3) assumes that the time mitigation factors have sequential impacts. In a case where concurrent impact is assumed, the minimum value of the mitigation factors (rather than the product of the factors) is considered.

Table 2 presents the considered recovery processes and corresponding time for community-managed and agency-managed (re)construction projects. The adopted times are based on outputs from semi-structured interviews with NGOs, engineering firms, and contractors actively involved in reconstruction projects in the Palu region[48].

Supplementary Table 3 presents a range of recovery time mitigation and amplification factors based on a survey[40] of observations, interviews, and focus group discussions from published studies that have compared pre- and post-disaster reconstruction projects in lower-middle income countries. For example, reports[64] show that the absence of local government resulted in several months of delays before post-disaster emergency support could reach specific regions affected by the 2015 Gorkha earthquake in Nepal. Furthermore, resolving fundamental issues (e.g., tender process, building permits, and land acquisition) took at least four times the required time. Data[55] show that the recovery rate of buildings in Sri Lanka following the 2004 Indian Ocean tsunami was eight times larger in conflict zones compared to zones outside the conflict region. Reports[65,66] have highlighted that poor management skills can lead to construction delays, rejection by beneficiaries, rework, and demolition of newly constructed buildings. Such delays can impede the recovery time by a factor of up to three. Furthermore, the current authors[48] interviewed engineers, contractors and NGO officials involved in the post-2018 Palu earthquake recovery. The interviews provided information on the increase in the time to complete various tasks relative to pre-disaster

scenario. All this information contributed to Supplementary Table 3 in the supplementary notes. A key limitation of the proposed recovery time mitigation and amplification factors (Supplementary Table 3) is that they are based on limited data. Future studies can refine these factors when more data become available. Also, we recognize that recovery-impeding factors (and related data) vary for different events, countries, and local contexts. This is a big challenge for data transferability. To address this, the factors presented in the table are provided as ranges based on values from different countries of similar human development indices, rather than single average or median values. It is intended that users can select a value within the range that best replicates their own local context and adopt it in the PERT model. The PERT model further allows users to simulate desired levels of pessimism in recovery time analyses. It is expected that risk modeling tools adopting this approach (including selecting relevant factors for their own scenario) will provide useful insights into possible future scenarios and their outcomes.

Equation (3) accounts for the fact that the time spent in each recovery phase is dependent on the functionality level of the building. $T_{int}$ for constructing a new TLC would be different from the permanent construction of a new building to replace an FL3 building or repair an FL2 building. As earlier mentioned, although not shown in Eq. (3), the functionality level of each building is conditioned on the IM realization.

Similarly, the pessimistic time $b$ (i.e., the maximum time to complete each phase) is estimated as:

$$b_{i,z}(t)|STEEP,FL_z = \prod_{n=1}^{q} \beta_{i,n}(t)|STEEP \times T_{i,z}|FL_z \quad (4)$$

where $n$ are the considered amplification factors ($\beta$) ($n = 1, 2, ..., q$) influencing phase $i$. Similarly, $b_{i,z}$ is time-dependent. Equation (4) assumes that the time amplification factors have sequential impacts. In a case where concurrent impact is assumed, the maximum value of the amplification factors (rather than the product of the factors) is used.

The most likely duration ($m$) captures the highest likelihood of completing a recovery phase in a given timeframe. If there is a higher likelihood that the mitigation factors would be more prevalent than the amplification factors, $m$ is defined to be closer to $a$. Otherwise, $m$ is defined closer to $b$. When uncertain, $m$ can be defined as $0.5(a+b)$.

The defined time parameters (i.e., $a$, $m$, and $b$) are then used to generate a PERT distribution for each recovery phase. The defined probabilistic duration parameters for each task are then used to carry out plain Monte Carlo sampling. Hence, for each realization $j$ of IM, the probable recovery time for a school building $z$ to achieve its full functionality $Q_{z,full}$ (either through a TLC or permanent building construction) is estimated as:

$$t^z_{r,j}|STEEP,FL_z = T^z_{insp,j}|STEEP,FL_z + T^z_{mob,j}|STEEP,FL_z + T^z_{int,j}|STEEP,FL_z \quad (5)$$

The post-earthquake recovery trajectory for an entire school with $n_{bld}$ buildings is defined as:

$$Q^s_j(t) = \sum_{z=1}^{n_{bld,s}} Q^{z,s}_j(t) \quad (6)$$

The recovery time for a school is defined as:

$$t^s_{r,j} = \max\left(t^1_{r,j}, t^2_{r,j}, \ldots, t^{n_{bld,s}}_{r,j}\right) \quad (7)$$

The recovery trajectory for the entire community with $n_s$ schools for each IM realization $j$ is defined as:

$$Q^c_j(t) = \sum_{s=1}^{n_s} \sum_{z=1}^{n_{bld,s}} Q^{z,s}_j(t) \quad (8)$$

The recovery time for the entire community is defined as:

$$t_{R,j} = \max\left(t_{r,j}^1, t_{r,j}^2, \ldots, t_{r,j}^s\right) \qquad (9)$$

## Data availability

The school database used in this study is described in Opabola et al.[42] and is publicly available[37]. For confidentiality purposes and compliance with the guiding ethics policy, any information that can be used to identify the schools has been redacted in the uploaded database. However, we are willing to share more information (under strict protocols) with other researchers. The stakeholder engagement, including guiding questions, adopted in the study are reported in Opabola et al.[48].

## Code availability

All code used to conduct this analysis is freely available at https://github.com/TayoOpabola/Post-disaster-modelling-of-school-infrastructure (https://doi.org/10.5281/zenodo.10070853).

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

## Acknowledgements
The study was carried out through the MultiVERSE project, funded by the UK Research and Innovation (UKRI) with project reference EP/X023710/1.

## Author contributions
E.O.: Conceptualization; Data curation; Formal Analysis; Methodology; Writing—original draft; Writing—review & editing. C.G.: Conceptualization; Writing—original draft; Writing—review & editing.

## Competing interests
The authors declare no competing interests.
