## [Peer Review File · Nature Communications]

Informing disaster-risk management policies for education infrastructure using scenario-based recovery analysesREVIEWER COMMENTS

Reviewer #1 (Remarks to the Author):

The authors present an interesting study on the post-earthquake recovery of affected schools with focus on Indonesia. They use risk analysis techniques to evaluate how long it takes to recover the functionality of schools under different response mechanisms. I have the following four comments.

1. It is hard for the reader to grasp what is state-of-the-art on the topic and what are the methodological contributions of the paper. In my opinion, there are already many methodologies with risk-based assessments of the recovery of buildings. The paper would benefit from a clearer and more thorough discussion on what is new and whether and how the proposed framework is generalizable to multiple regions/hazards. There are other studies on schools that have already made important contributions on modeling recovery of schools at the regional level, including nice optimization algorithms for decision-making on school repairs. While some are mentioned (e.g., Alisjahbana et al. , Hassan et al.), not enough discussions and descriptions are provided.

There are other several studies, including but not limited to earthquakes, that this study could review to better frame the state of art and explain the paper's contributions. Here are some examples, but there are many.

- S. S. Schulze, E. C. Fischer, S. Hamideh, and H. Mahmoud, "Wildfire impacts on schools and hospitals following the 2018 California Camp Fire," *Nat. Hazards*, vol. 104, no. 1, pp. 901–925, 2020, doi: 10.1007/s11069-020-04197-0.

- M Hosseini, YO Izadkhah. (2006). Earthquake disaster risk management planning in schools. *Disaster Prevention and Management*.

- T C Mutch (2015) The role of schools in disaster settings: Learning from the 2010–2011 New Zealand earthquakes. *International Journal of Educational Development*

- B. S. Lai, A. M. Esnard, C. Wyczalkowski, R. Savage, and H. Shah, "Trajectories of School Recovery After a Natural Disaster: Risk and Protective Factors," *Risk, Hazards Cris. Public Policy*, vol. 10, no. 1, pp. 32–51, 2019, doi: 10.1002/rhc3.12158.

The authors could also refer and compare their methodologies to the ones that have been proposed for other critical infrastructure to better explain their contributions. For example,

- EM Hassan, H Mahmoud (2019) Full functionality and recovery assessment framework for a hospital subjected to a scenario earthquake event. *Engineering Structures*

- I Alisjahbana, L Ceferino, A Kiremidjian. (2022) Prioritized reconstruction of healthcare facilities after earthquakes based on recovery of emergency services. *Risk analysis*

2. What are the main limitations of the framework? And what future research is this study encouraging? For example, the authors explore three finance mechanisms. Are they equally confident in their models for the three of them? Accounting for the influence of socio-political factors is hard because they are hard to predict, especially after disasters. Probably, the authors want to explain better the limitations of these analyses as well as other limitations in the models.

3. I am not sure that NGOs and communities are the main drivers of recovery of schools (line 241). The authors cite reference 13, which studies Nepal. As NGOs are a central component of the main conclusions of the paper, it would be good to clarify whether the results are generalizable to other regions or only valid for Nepal. In many other regions (including the Global South), the government and the private sector (e.g., engineering firms) lead the recovery.

4. What about the relocation of kids to different schools? If one is closed, I imagine the kids will go to the next closest school, as suggested in the paper below. Would accounting for this make your results different? I think this point is also part of my comment 1, where the generalizability of the results (in terms of financial mechanisms, regions, policies, and natural hazards) needs to be better explained.

- Alisjahbana, A. Graur, I. Lo, and A. Kiremidjian, "Optimizing strategies for post-disaster reconstruction of school systems," *Reliab. Eng. Syst. Saf.*, vol. 219, no. October 2021, p. 108253, 2022, doi: 10.1016/j.res.2021.108253.

Reviewer #2 (Remarks to the Author):

Review of: Informing disaster-risk management policies for education infrastructure using scenario-based recovery analyses

Authors: Eyitayo Opabola, Carmine Galasso

I enjoyed reading this work. This work covers an important topic in post-disaster management, specifically school continuity following a disruption from natural hazard extreme event. The paper further proposes a model to predict delays in school operations due to various recovery obstacles, and demonstrates its use in a semi-hypothetical example of schools in Silawesi, Indonesia.

Overall, the work covers an important topic and proposes a useful tool to support risk-informed decision-making for minimizing delays in schooling due to natural hazards. However, numerous important assumptions are taken for granted with little explanation or supporting literature. It is further difficult to draw the lines between collected information (presumably an extensive survey and focus group discussion with school officials) to the specific model assumptions in terms of delay time, amplification and mitigation factors etc. It is the belief of this reviewer that the paper would be strengthened through the inclusion of the following considerations:

General comments:

Importance of school continuity: The work would be strengthened by more extensive motivation. It takes for granted that school continuity is important. However, why should it receive particular attention and separate modeling? Only one sentence alludes to "available evidence that out-of-school children are susceptible to various forms of exploitation." The importance of schooling continuity is a well researched topic, and the paper should explore this in much more depth. For instance it has been shown that the impact of school discontinuity can last decades following an event (research into impacts of disaster on social capital).

Early financing as a critical obstacle to education continuity: The model indicates that access to early financing is the main challenge to education continuity following disaster. This is presumably obtained from the focus-group discussions. However if this is indeed true of the Silawesi recovery process, generalizing this statement is not obvious. In many other disasters, financing is rarely the obstacle to recovery, especially for schools since they are often prioritized both by national governments and by the international aid institutions. Typically issues of access to construction equipment, labor, difficulty of the specific site, permitting in risk-prone areas, access to expertise, (etc) are controlling issues in reconstruction. At minimum it should be specified precisely how access to early financing would by itself enable early recovery of schools. There is much research on obstacles to recovery, and the paper would benefit from more engagement with that.

Distinction between NGO-managed and community-managed school recovery: the paper makes a clear distinction between NGO-managed and community-managed school recovery work. It is quite unclear what this distinction means in reality, and where other interventions fit in this duality? Are government-led initiatives assumed to be 'community-managed'? How about foreign-aid funded but locally implemented interventions? How about local community NGOs? This binary classification is reductive, and likely not the most essential in understanding recovery delays.

NGO-managed recovery is delayed: Separately, there is a very rich literature on community vs agency-driven reconstruction programs, non of which is cited in this work (though much of this literature focuses on housing projects specifically, and it is unclear how that would generalize to school or other infrastructure). The paper starts with the premise that NGO-led recovery is delayed, which may be true but would need much more extensive referencing. What makes NGO-led recovery delayed? Why would community-managed recovery not be delayed? Are these delays intrinsic to the projects being NGO-led, or is it something about the way NGOs operate, or the type of projects they select?

The paper assumes that NGO-led recovery is delayed, and the model then confirms this delay, but it seems to be a circular argument, and the conclusions are a direct reflection of this first assumption. The paper would therefore benefit from explaining the added information provided by

the model.

Methods: The methods section does not explain how the recovery times were obtained. The paper makes references to focus-group discussions, but it is unclear how these are transformed into assumptions for delay-times, amplification and mitigation factors etc. Likewise figure 6: where is this data from?

Data accessibility: Suggest making the data, or sample data and sample analysis available, so other researchers can more easily use and reference your proposed model.

Figure 5: Unclear how to read figure 5. The way I read it it would seem that NGO-managed programs are in fact much faster to reconstruct than community-managed ones. Are the curves in the legend switched? Also is there data underlying these curves, or are these subjectively assigned? If there is data, should be included in the plot.

References: several of the references are from grey/news literature and could be replaced with peer-reviewed papers. Some references are missing journals or sources.

Detailed comments:

Please see annotations directly in text. These were done in a moving vehicle, so apologies for shaky hands.

Response to comments for “Informing disaster-risk management policies for education infrastructure using scenario-based recovery analyses” by Eytayo Opubola and Carmine Galasso

NCOMMS-22-42189-T

We thank the reviewers for their thoughtful and insightful comments, which have significantly improved the quality of the revised manuscript. The reviewer’s comments have been numbered and listed below (in *italic*), followed by our responses in **blue**. Extracts from the manuscript are reported in **red**, with additions to the revised manuscript in **bold**.

Reviewer #2

I enjoyed reading this work. This work covers an important topic in post-disaster management, specifically school continuity following a disruption from natural hazard extreme event. The paper further proposes a model to predict delays in school operations due to various recovery obstacles, and demonstrates its use in a semi-hypothetical example of schools in Silawesi, Indonesia.

Overall, the work covers an important topic and proposes a useful tool to support risk-informed decision-making for minimizing delays in schooling due to natural hazards.

However, numerous important assumptions are taken for granted with little explanation or supporting literature. It is further difficult to draw the lines between collected information (presumably an extensive survey and focus group discussion with school officials) to the specific model assumptions in terms of delay time, amplification and mitigation factors etc. It is the belief of this reviewer that the paper would be strengthened through the inclusion of the following considerations:

We would like to thank the reviewer for the positive overall assessment of our contribution in terms of importance and value. We appreciate the reviewer’s comments in terms of further clarifying specific study assumptions and have made various changes to the manuscript’s content (described in detail in the comments below) in response to those considerations.

Importance of school continuity: *The work would be strengthened by more extensive motivation. It takes for granted that school continuity is important. However, why should it receive particular attention and separate modeling? Only one sentences alludes to “available evidence that out-of-school children are susceptible to various forms of exploitation.” The importance of schooling continuity is a well researched topic, and the paper should explore this in much more depth. For instance it has been shown that the impact of school discontinuity can last decades following an event (research into impacts of disaster on social capital).*

We agree with the reviewer’s point on providing more background statements on the importance of school continuity. To reflect this, the following modifications have been made to line 48:

“Several studies have highlighted the importance of post-disaster school continuity. There are various unintended social and economic consequences of education disruption to schools, students, teachers, their families, and the community at large. For example, evidence (Sanderson and Ramalingam 2015) shows that out-of-school children are susceptible to various forms of exploitation (including child labor) and violence (especially in temporary camps), with severe effects on children’s long-term development. **In addition, it has been reported that education disruption in school children may lead to long-term poorer physical and mental health, leading to a loss of productivity and earnings in adulthood** (Andrabi et al. 2021; Baez et al. 2010). **The socioeconomic conditions of staff of closed schools may also be negatively impacted if they need to find alternative jobs (in an already chaotic post-disaster situation) to make ends meet. Also, parents (and carers) may have to spend time off work to take care of their children, resulting in a significant productivity loss for the economy** (Chen et al. 2011) **as well as well-being losses** (Markhvida et al. 2020) **for them. For nations, the combined influence of education disruption on school children, staff, and their**

families may result in up to a 6% loss in future gross domestic product (Hanushek and Woessmann 2020; Sadique et al. 2008; Sander et al. 2009). Hence, post-disaster education continuity must be a priority for any nation.”

***Early financing as a critical obstacle to education continuity:** The model indicates that access to early financing is the main challenge to education continuity following disaster. This is presumably obtained from the focus-group discussions. However if this is indeed true of the Silawesi recovery process, generalizing this statement is not obvious. In many other disasters, financing is rarely the obstacle to recovery, especially for schools since they are often prioritized both by national governments and by the international aid institutions. Typically issues of access to construction equipment, labor, difficulty of the specific site, permitting in risk-prone areas, access to expertise, (etc) are controlling issues in reconstruction. At minimum it should be specified precisely how access to early financing would by itself enable early recovery of schools. There is much research on obstacles to recovery, and the paper would benefit from more engagement with that.*

Thanks for this very important comment. Our study does not aim to portray lack of early financing as ‘the main challenge to education continuity’. We apologize if this was the impression we gave. The intent of the analysis on early financing availability for constructing temporary learning centers is to show how early financing can positively influence education continuity. We agree with the reviewer that the government prioritizes reconstructing permanent school structures. However, it is important to note that because they are ‘makeshift’ structures (i.e., tents), temporary learning centers are typically non-engineered, use readily available local materials, and do not require skilled labor or heavy equipment for their construction. Hence, the technical factors that affect the construction of permanent structures (where building permits are needed, the tender process may be necessary, and the lack of building materials and technical know-how are more influential) are not significant for temporary learning centers. Nevertheless, we note that the proposed methodology can capture the influence of other recovery-impeding factors on the construction of temporary learning centers. Indeed, we discussed the relevant recovery-impeding factors in lines 85-94. The analyses in Lines 385-413 showcase how technical, socioeconomic, and political factors can significantly influence the reconstruction of permanent school structures.

The following modifications have been made to the paper to address the reviewer’s comment and avoid confusion:

Line 277 has been modified as:

“Using the proposed framework, we demonstrate how local authorities can estimate a given school’s education continuity recovery trajectory (through the construction of TLCs), accounting for the uncertainties involved in the process. We assumed that because they are ‘makeshift’ structures (i.e., tents), TLCs are typically non-engineered, use local readily available materials, and do not require skilled labor or heavy equipment for their construction. Hence, the technical factors that affect the construction of permanent structures (where building permits are needed, the tender process may be necessary, and the lack of building materials and technical know-how are more influential) are not significant for TLCs. Although not considered in this analysis, we note that the proposed methodology can capture the influence of other recovery-impeding factors on the construction of TLCs.”

Line 294 has been modified as:

“For the case-study community, we show that sufficient anticipatory funding can reduce the recovery time for education continuity (through rapid construction of TLCs) by a factor of up to three.”

Also, based on a related comment by Reviewer #1, the following additions have been made to Line 80:

“In comparison with studies on other infrastructure systems (e.g., residential and business building stock (Burton et al. 2016), hospitals (Hassan and Mahmoud 2019), and utility networks (Unnikrishnan and van de

Lindt 2016), fewer research studies have been carried out on post-disaster recovery of physical education infrastructures. **We also note that because building functionality heavily depends on occupancy type, post-disaster recovery modeling frameworks for other infrastructure systems/occupancies cannot be generalized.**

On the qualitative side, various resilience-enhancing policies for school infrastructures have been proposed by different studies in recent years (Hosseini and Izadkhah 2006; Matyas and Pelling 2015). Quantitative studies on the resilience of physical education infrastructures started gaining widespread attention in recent years. Available quantitative studies are either field-data-based or simulation-based. Field-based studies (Eghbali et al. 2020; Ghafory-Ashtiany and Hosseini 2008; Platt et al. 2020; Westoby et al. 2021) have emphasized the prolonged post-disaster school reconstruction process and its negative influence on education continuity. The former has been attributed to funding delays, contract issues, the use of schools as temporary shelters by local communities, the political setting, land acquisition issues (in cases where school relocation is needed), inaccessible roads for transporting materials to remote locations, management type (i.e., community-managed or agency-managed projects), lack of skilled labor, and flawed planning processes.

Studies have proposed simulation-based probabilistic frameworks to simulate the post-disaster resilience of education systems. **Some of these studies (Aghababaei and Koliou 2022; Hassan et al. 2020) have developed probabilistic frameworks to simulate the post-disaster recovery trajectory of school infrastructure. However, these studies do not consider the influence of the previously mentioned sociocultural, political, economic, environmental, and technical factors that significantly influence the recovery trajectory of schools in developing/marginalized communities. This may be attributed to the fact that their studies are developed to target developed communities (e.g., the USA).** Hence, the applicability of such frameworks to tackle a wide range of multi-dimensional issues within a marginalized **community** may be limited. “

***Distinction between NGO-managed and community-managed school recovery:** the paper makes a clear distinction between NGO-managed and community-managed school recovery work. It is quite unclear what this distinction means in reality, and where other interventions fit in this duality? Are government-led initiatives assumed to be ‘community-managed’? How about foreign-aid funded but locally implemented interventions? How about local community NGOs? This binary classification is reductive, and likely not the most essential in understanding recovery delays.*

We thank the reviewer for this comment. We highlight that there is a difference between funding sources and project management mechanisms. For example, a project could be funded through foreign aid and managed by the community. Likewise, a project could be financed and managed through a foreign agency. A foreign-aid-funded but locally implemented intervention is a community-managed project. Based on the reviewer’s comment, we agree that further clarification (and appropriate definition) needs to be provided here. Also, to avoid confusion, the term NGO-managed has been modified to ‘agency-managed’. The following clarifications have been added to Lines 305 to 313:

“The government, NGOs (both local and international), or the host communities typically provide post-disaster school reconstruction project funding. However, the project management style varies. Post-disaster recovery reports (Westoby et al. 2021) in major global disasters show that most permanent school-building reconstruction projects are either community-managed or agency-managed. The community-managed approach is a locally implemented approach where the construction management (e.g., material acquisition, selection of building contractors) is led by local authorities, school management committees or host community leaders. The agency-managed approach is a case where a government or non-government entity leads the construction management.”

***NGO-managed recovery is delayed:** Separately, there is a very rich literature on community vs agency-driven reconstruction programs, non of which is sited in this work (though much of this litterature focuses on housing projects specifically, and it is unclear how that would generalize to school or other infrastructure). The paper*

starts with the premise that NGO-led recovery is delayed, which may be true but would need much more extensive referencing. What makes NGO-led recovery delayed? Why would community-managed recovery not be delayed? Are these delays intrinsic to the projects being NGO-led, or is it something about the way NGOs operate, or the type of projects they select?

As the reviewer pointed out, most existing literature on community vs. agency-driven reconstruction projects focuses on residential buildings. Therefore, we avoided populating our paper (which focuses on schools) with references to residential building-related studies to prevent confusion. Also, as the reviewer highlights, it is unclear that findings related to residential buildings could be generalized to school buildings or other infrastructure.

The following sentences have been provided in Line 314 to discuss the conclusions of the literature on community vs agency-driven projects:

“Several field-based comparative studies (Andrew et al. 2013; Elkahout 2019; Karunasena and Rameezdeen 2010; Opabola et al. 2022; Ratnayake and Rameezdeen 2008; Westoby et al. 2021) have highlighted the successes and weaknesses of community-managed and agency-managed reconstruction projects (residential and school buildings) in developing countries. For example, the abundance of local knowledge and experience positively influences the recovery process of community-managed projects and achieves higher beneficiary satisfaction. Also, the fact that the drivers (i.e., local authorities, school management committees or host community leaders) of community-managed projects typically reside in the communities is a strong motivation to ensure rapidity and higher quality in reconstruction projects. On the other hand, the agency-managed approach has often been described as a ‘one-size-fits-all’ approach intended to suit donors and implementing agencies and rarely involve the target beneficiaries (Andrew et al. 2013; Elkahout 2019). Typically, agency-managed projects are subjected to delays resulting from internal bureaucracy, agreements with the local authority, community, and school authorities, and the lengthy bidding process for engineers and contractors.” We note that principal interviews and focus group discussions with key stakeholders (engineers, contractors, government, and NGO officials)(Opabola et al. 2022) actively involved in the recovery process of the post-2018 Palu earthquake highlighted significant delays of agency-managed projects in Palu.”

The paper assumes that NGO-led recovery is delayed, and the model then confirms this delay, but it seems to be a circular argument, and the conclusions are a direct reflection of this first assumption. The paper would therefore benefit from explaining the added information provided by the model.

As discussed in our paper, the developed framework is a simulation tool that local authorities can use to make risk-informed decisions. The case-study scenarios in our paper mainly showcase the framework’s capabilities to effectively model lessons learned from the field. It translates those lessons/findings into modeling assumptions that enable a proper quantification of school recovery trajectories under those assumptions. We note that the framework can also capture scenarios where agency-managed projects may have shorter recovery times than community-managed projects.

The delays we considered in the study are based on outcomes from principal interviews and focus group discussions with key stakeholders (engineers, contractors, government, and NGO officials) actively involved in the recovery process of the post-2018 Palu earthquake.

To better reflect these comments, we added the following sentence to Line 326:

“We note that principal interviews and focus group discussions with key stakeholders (engineers, contractors, government, and NGO officials) (Opabola et al. 2023) actively involved in the recovery process of the post-2018 Palu earthquake highlighted significant delays of agency-managed projects in Palu.”

Methods: The methods section does not explain how the recovery times were obtained. The paper makes references to focus-group discussions, but it is unclear how these are transformed into assumptions for delay-times, amplification and mitigation factors etc. Likewise figure 6: where is this data from?

We thank the reviewer for highlighting the need for more clarity on the recovery time modeling framework. Equations (3) to (5) present how recovery times are estimated. We realized that we did not explicitly mention the sources for the typical average time to carry out mobilization and intervention tasks in pre-disaster scenarios. The average recovery time is discussed in Lines 341-345.

“In a pre-disaster scenario, the required time to construct a typical **one-story** school building in Palu is four to six months (120 to 180 days). **Table 1 presents the considered recovery phases and corresponding time for community-managed and agency-managed projects. The adopted times are based on outputs from semi-structured interviews with NGOs, engineering firms, and contractors actively involved in reconstruction projects in the Palu region. By considering recovery-impeding factors (material procurement and delays in agreements, tender process, and permits), the CDF for community-managed projects is shown in Figure 5a. Community-managed projects, on average, would be completed two times slower than in a pre-disaster scenario.**

Table 1 – Average time for various stages of the recovery process for one-story school buildings in Central Sulawesi. These values are from semi-structured interviews with NGOs, engineering firms, and contractors actively involved in reconstruction projects in the Palu region

Recovery phase	Average time T [days]
Agreement between agencies and local authorities*	90
Tender process*	180
Structural design	60
Building construction permit	90
Reconstruction time (includes substructure and superstructure)	180

***Only considered for the agency-managed projects”**

A range of amplification and mitigation factors based on an extensive literature survey by the authors in a separate study has now been added to the Method section (See below).

Table 4 – Time amplification and mitigation factors for recovery time modeling. Each range of factors has been developed through a literature survey (Opabola and Galasso n.d.) of observations, interviews, and focus group discussions carried out by other authors who have compared pre- and post-disaster reconstruction projects in developing countries.

Parameter	Factor
Time amplification factors	
Land dispute resolution	1.25 – 2
Pandemic	1.5 – 3
Delay in material procurement	1.2 – 2.5
Hostile political conditions	1.5 – 5
Poor management skills	1.5 – 3
Funds disbursement	1.25 – 3
Technical delays	1.2 – 2.5
Time mitigation factor	
Voluntary mobilization	0.5 – 0.9

The following clarification has been made to the caption of figure 6.

“Figure 6 – Median post-disaster reconstruction time for school buildings under (1) hostile political conditions; (2) pandemic; (3) land dispute; (4) poor management skills of contractors; (5) a combination of land dispute and poor management skills of contractors. **The time amplification factors used in simulating the influence of these conditions in Eqs (3)-(5) are based on Table 5 in Methods. For each of the five scenarios, the optimistic time (a) was estimated using a time mitigation factor of 1.0, while the pessimistic time (b) is based on the upper-bound values of the time amplification factors in Table 5.**”

Data accessibility: *Suggest making the data or sample data and sample analysis available, so other researchers can more easily use and reference your proposed model.*

All adopted codes and related input data have now been uploaded to GitHub. The link is provided below:

<https://github.com/TayoOpabola/Post-disaster-modelling-of-school-infrastructure>

Figure 5: *Unclear how to read figure 5. The way I read it it would seem that NGO-managed programs are in fact much faster to reconstruct than community-managed ones. Are the curves in the legend switched? Also is there data underlying these curves, or are these subjectively assigned? If there is data, should be included in the plot.*

Many thanks for pointing this out. Indeed, there was a typo in the legend of Figure 5 in the original manuscript; apologies for this. The typo has now been fixed.

We have also provided more discussions on the plot in the caption of Figure 5:

“Figure 5 – **Simulated** cumulative distribution function (CDF) for the reconstruction time of community-managed and NGO-managed school building reconstruction in the testbed community **developed using the recovery time model presented in Equations (3) – (5). The CDF accounts for recovery-impeding (i.e., time amplification) factors that may influence both community and agency-managed projects. The community-managed school building projects have a lower median recovery time because of little to no internal bureaucratic issues, more coordinated arrangement with relevant stakeholders, and little to no delay in a contract bidding process in these types of projects.**”

The following sentences have been added to Line 335:

“We explicitly account for the influence of delayed bidding process, internal bureaucracy on the bidding and construction mobilization and construction time of agency-managed projects through recovery time amplification factors described in Methods. In line with the earlier discussions, these factors are assumed to have a lesser impact on community-managed projects and are not considered.”

We would also like to thank the reviewer for all the editorial comments provided in the PDF copy of our manuscript. All the editorial comments have now been addressed.

Reviewer #1

The authors present an interesting study on the post-earthquake recovery of affected schools with focus on Indonesia. They use risk analysis techniques to evaluate how long it takes to recover the functionality of schools under different response mechanisms. I have the following four comments.

We would like to thank the reviewer for the positive overall assessment of our contribution in terms of interest. We appreciate the reviewer's comments in terms of further clarifying specific study assumptions and have made various changes to the manuscript's content (described in detail in the comments below) in response to those considerations.

It is hard for the reader to grasp what is state-of-the-art on the topic and what are the methodological contributions of the paper. In my opinion, there are already many methodologies with risk-based assessments of the recovery of buildings. The paper would benefit from a clearer and more thorough discussion on what is new and whether and how the proposed framework is generalizable to multiple regions/hazards. There are other studies on schools that have already made important contributions on modeling recovery of schools at the regional level, including nice optimization algorithms for decision-making on school repairs. While some are mentioned (e.g., Alisjahbana et al. , Hassan et al.), not enough discussions and descriptions are provided. There are other several studies, including but not limited to earthquakes, that this study could review to better frame the state of art and explain the paper's contributions. Here are some examples, but there are many.

- S. S. Schulze, E. C. Fischer, S. Hamideh, and H. Mahmoud, "Wildfire impacts on schools and hospitals following the 2018 California Camp Fire," *Nat. Hazards*, vol. 104, no. 1, pp. 901–925, 2020, doi: 10.1007/s11069-020-04197-0.

- M Hosseini, YO Izadkhah. (2006). *Earthquake disaster risk management planning in schools. Disaster Prevention and Management.*

- T C Mutch (2015) *The role of schools in disaster settings: Learning from the 2010–2011 New Zealand earthquakes. International Journal of Educational Development.*

- B. S. Lai, A. M. Esnard, C. Wyczalkowski, R. Savage, and H. Shah, "Trajectories of School Recovery After a Natural Disaster: Risk and Protective Factors," *Risk, Hazards Cris. Public Policy*, vol. 10, no. 1, pp. 32–51, 2019, doi: 10.1002/rhc3.12158.

The authors could also refer and compare their methodologies to the ones that have been proposed for other critical infrastructure to better explain their contributions. For example,

- EM Hassan, H Mahmoud (2019) *Full functionality and recovery assessment framework for a hospital subjected to a scenario earthquake event. Engineering Structures*

- I Alisjahbana, L Ceferino, A Kiremidjian. (2022) *Prioritized reconstruction of healthcare facilities after earthquakes based on recovery of emergency services. Risk analysis*

Thanks for this comment. As pointed out by the reviewer, we explicitly mentioned some of the studies that specifically focused on school buildings and earthquake hazard in low-income countries. However, we intentionally did not mention studies on residential buildings. In fact, while post-disaster damage state definitions are generally not peculiar to occupancy type, the description of post-disaster functionality strongly depends on occupancy type (among other "local" and "hazard-specific" factors). Regarding the key gap we identified in existing studies, in the same paragraph (i.e., Lines 95-104), we briefly summarised that most existing studies do not explicitly consider the influence of the sociocultural, political, economic, environmental, and technical factors on the recovery trajectory in their proposed frameworks. This is what our study addresses.

The following modifications have been made to Line 80 to clarify those aspects better and reflect the reviewer's comments:

“In comparison with studies on other infrastructure systems (e.g., residential and business building stock (Burton et al. 2016), hospitals (Hassan and Mahmoud 2019), and utility networks (Unnikrishnan and van de Lindt 2016), fewer research studies have been carried out on post-disaster recovery of physical education infrastructures. We also note that because building functionality heavily depends on occupancy type, post-disaster recovery modeling frameworks for other infrastructure systems/occupancies cannot be generalized.

On the qualitative side, various resilience-enhancing policies for school infrastructures have been proposed by different studies in recent years (Hosseini and Izadkhah 2006; Matyas and Pelling 2015). Quantitative studies on the resilience of physical education infrastructures started gaining widespread attention in recent years. Available quantitative studies are either field-data-based or simulation-based. Field-based studies (Eghbali et al. 2020; Ghafory-Ashtiany and Hosseini 2008; Platt et al. 2020; Westoby et al. 2021) have emphasized the prolonged post-disaster school reconstruction process and its negative influence on education continuity. The former has been attributed to funding delays, contract issues, the use of schools as temporary shelters by local communities, the political setting, land acquisition issues (in cases where school relocation is needed), inaccessible roads for transporting materials to remote locations, management type (i.e., community-managed or agency-managed projects), lack of skilled labor, and flawed planning processes.

Studies have proposed simulation-based probabilistic frameworks to simulate the post-disaster resilience of education systems. **Some of these studies (Aghababaei and Koliou 2022; Hassan et al. 2020) have developed probabilistic frameworks to simulate the post-disaster recovery trajectory of school infrastructure. However, these studies do not consider the influence of the previously mentioned sociocultural, political, economic, environmental, and technical factors that significantly influence the recovery trajectory of schools in developing/marginalized communities. This may be attributed to the fact that their studies are developed to target developed communities (e.g., USA). Hence, the applicability of such frameworks to tackle a wide range of multi-dimensional issues within a marginalized community may be limited.**

Fewer studies have sought to contribute to post-disaster resilience modeling in the developing world. For example, Alisjahbana et al. (2021) developed an optimization approach for school reconstruction scheduling by minimizing the sum of the distance all students in the region have to travel until all schools in the region are reconstructed. However, it is noted that this approach's applicability is limited when critical issues such as land availability, construction site accessibility, and school level can influence reconstruction projects. Hence, a multicriteria decision-making approach that accounts for the influence of various factors on school reconstruction prioritization is needed.”

2. What are the main limitations of the framework? And what future research is this study encouraging? For example, the authors explore three finance mechanisms. Are they equally confident in their models for the three of them? Accounting for the influence of socio-political factors is hard because they are hard to predict, especially after disasters. Probably, the authors want to explain better the limitations of these analyses as well as other limitations in the models.

The framework has been proposed as a tool for various end users, including local authorities, to make risk-informed planning decisions. These local authorities are expected to be conversant with the economic and sociopolitical factors that can affect their communities. The recovery mitigating factors, which have been calibrated to other past events, provide a good range of values that modelers can use to understand the sensitivity of a community to various possible economic and sociopolitical factors in their communities from a *what-if* perspective. Of course, additional data will enable us to better calibrate these recovery time mitigating factors.

What we believe is a key limitation of the framework is the fact that we focus on physical infrastructure when it is known that non-physical aspects of recovery (e.g., psychosocial) may affect education continuity.

We added the following sentence in Line 63:

“There are two distinct domains of post-disaster school recovery necessary for education continuity – physical and non-physical domains. The physical domain of post-disaster recovery is related to the conditions of the physical infrastructure (e.g., classrooms, laboratories, water, sanitation and hygiene (WASH) facilities, power, and water utilities). For instance, the non-physical domain is associated with the post-disaster management structure and psychosocial recovery of school children and staff (Kronenberg et al. 2010; Nastasi et al. 2011). We note that our study focuses on the analytical modeling of the physical domain of school recovery.”

We further buttressed this limitation in Line 191:

“One of the limitations of this study is the focus on the physical domain of recovery. Education continuity and recovery trajectory of schools may be affected by psychosocial factors that are not explicitly modeled in this study.”

The following sentence has also been added to Line 610:

“A key limitation of the proposed recovery time mitigation and amplification factors (Table 4) is the fact that they are based on limited data. Future studies can refine these factors when more data become available.”

3. I am not sure that NGOs and communities are the main drivers of recovery of schools (line 241). The authors cite reference 13, which studies Nepal. As NGOs are a central component of the main conclusions of the paper, it would be good to clarify whether the results are generalizable to other regions or only valid for Nepal. In many other regions (including the Global South), the government and the private sector (e.g., engineering firms) lead the recovery.

We thank the reviewer for this comment. We agree that the initial discussions in the paper do not clearly explain what we mean by community-managed and agency-managed projects. Therefore, further clarifications have been provided from Lines 306 to 323:

“The government, NGOs (local and international), or the host communities typically provide post-disaster school reconstruction project funding. However, the project management style varies. Post-disaster recovery reports (Westoby et al. 2021) in major global disasters show that most permanent school-building reconstruction projects are either community-managed or agency-managed. The community-managed approach is a locally implemented approach where the construction management (e.g., material acquisition, selection of building contractors) is led by local authorities, school management committees or host community leaders. The agency-managed approach is a case where a government or non-government entity leads the construction management.

Several field-based comparative studies (Andrew et al. 2013; Elkahlout 2019; Karunasena and Rameezdeen 2010; Opabola et al. 2022; Ratnayake and Rameezdeen 2008; Westoby et al. 2021) have highlighted the successes and weaknesses of community-managed and agency-managed reconstruction projects (residential and school buildings) in developing countries. The abundance of local knowledge and experience positively influences the recovery process of community-managed projects and also achieves higher beneficiary satisfaction. Also, the fact that the drivers of community-managed projects (i.e., local authorities, school management committees, or host community leaders) typically reside in the communities is a strong motivation to ensure rapidity and higher quality in reconstruction projects. On the

other hand, the agency-managed approach has been described as a ‘one-size-fits-all’ approach intended to suit donors and implementing agencies and rarely involve the target beneficiaries (Andrew et al. 2013; Elkahlout 2019).”

4. What about the relocation of kids to different schools? If one is closed, I imagine the kids will go to the next closest school, as suggested in the paper below. Would accounting for this make your results different? I think this point is also part of my comment 1, where the generalizability of the results (in terms of financial mechanisms, regions, policies, and natural hazards) needs to be better explained.

- Alisjahbana, A. Graur, I. Lo, and A. Kiremidjian, “Optimizing strategies for post-disaster reconstruction of school systems,” *Reliab. Eng. Syst. Saf.*, vol. 219, no. October 2021, p. 108253, 2022, doi: 10.1016/j.ress.2021.108253.

As mentioned in Line 539, “Aside from the four considered criteria, decision-makers may consider others (e.g., proximity to neighboring schools with available space for new students, school management type - i.e., private- or government-owned).”

However, we do not consider proximity to a neighboring school in the study as interviews with school principals in our case-study region do suggest that most schools prefer to construct/remain in temporary learning centers rather than relocate to neighboring schools. Furthermore, various studies (Rumberger and Larson 1998; Astone and McLanahan 1994; Mantzicopoulos & Knutson, 2000) have suggested that mobility is problematic to students, families, and schools.

Rumberger, R. W., and Larson, K. A. “Student mobility and the increased risk of high school dropout,” *American journal of Education*, V. 107, No. 1, 1998, pp. 1–35.

Astone, N. M., and McLanahan, S. S. “Family structure, residential mobility, and school dropout: A research note,” *Demography*, V. 31, No. 4, 1994, pp. 575–84.

Mantzicopoulos, P., and Knutson, D. J. “Head Start children: School mobility and achievement in the early grades,” *The Journal of Educational Research*, V. 93, No. 5, 2000, pp. 305–11.

The following sentence has been added to Line 539:

“Aside from the four considered criteria, decision-makers may consider others (e.g., proximity to neighboring schools with available space for new students, school management type - i.e., private- or government-owned). However, we do not consider proximity to neighboring schools in this study because other studies have highlighted the negative impact of school mobility (e.g., lower academic achievement, reduced social interactions, and health and developmental problems) on school children (Rumberger and Larson 1998; Astone and McLanahan 1994; Mantzicopoulos & Knutson, 2000)”.

References

- Aghababaei, M., and Koliou, M. (2022). "An agent-based modeling approach for community resilience assessment accounting for system interdependencies: Application on education system." *Engineering Structures*, 255, 113889.
- Alisjahbana, I., Graur, A., Lo, I., and Kiremidjian, A. (2021). "Optimizing strategies for post-disaster reconstruction of school systems." *Reliability Engineering & System Safety*, Elsevier Ltd, 219(October 2021), 108253.
- Andrabi, T., Daniels, B., and Das, J. (2021). "Human capital accumulation and disasters: Evidence from the Pakistan earthquake of 2005." *Journal of Human Resources*, University of Wisconsin Press, 0520-10887R1.
- Andrew, S. A., Arlikatti, S., Long, L. C., and Kendra, J. M. (2013). "The effect of housing assistance arrangements on household recovery: an empirical test of donor-assisted and owner-driven approaches." *Journal of Housing and the Built Environment*, 28(1), 17–34.
- Baez, J., De la Fuente, A., and Santos, I. V. (2010). "Do natural disasters affect human capital? An assessment based on existing empirical evidence." IZA discussion paper.
- Burton, H. V., Deierlein, G., Lallemand, D., and Lin, T. (2016). "Framework for Incorporating Probabilistic Building Performance in the Assessment of Community Seismic Resilience." *Journal of Structural Engineering*, 142(8), 1–11.
- Chen, W.-C., Huang, A. S., Chuang, J.-H., Chiu, C.-C., and Kuo, H.-S. (2011). "Social and economic impact of school closure resulting from pandemic influenza A/H1N1." *Journal of Infection*, 62(3), 200–203.
- Eghbali, M., Samadian, D., Ghafory-Ashtiany, M., and Raissi Dehkordi, M. (2020). "Recovery and reconstruction of schools after M 7.3 Ezgeleh-Sarpole-Zahab earthquake; part II: Recovery process and resiliency calculation." *Soil Dynamics and Earthquake Engineering*, Elsevier Ltd, 139(July), 106327.
- Elkahlout, G. (2019). "Post-conflict housing reconstruction in the Gaza Strip." *International Journal of Housing Markets and Analysis*, 13(2), 317–330.
- Ghafory-Ashtiany, M., and Hosseini, M. (2008). "Post-Bam earthquake: Recovery and reconstruction." *Natural Hazards*, 44(2), 229–241.
- Hanushek, E. A., and Woessmann, L. (2020). "The economic impacts of learning losses." OECD.
- Hassan, E. M., and Mahmoud, H. (2019). "Full functionality and recovery assessment framework for a hospital subjected to a scenario earthquake event." *Engineering Structures*, Elsevier, 188, 165–177.
- Hassan, E. M., Mahmoud, H. N., and Ellingwood, B. R. (2020). "Resilience of School Systems Following Severe Earthquakes." *Earth's Future*, 8(10).
- Hosseini, M., and Izadkhah, Y. O. (2006). "Earthquake disaster risk management planning in schools." *Disaster Prevention and Management: An International Journal*, 15(4), 649–661.
- Karunasena, G., and Rameezdeen, R. (2010). "Post-disaster housing reconstruction." *International Journal of Disaster Resilience in the Built Environment*, 1(2), 173–191.
- Kronenberg, M. E., Hansel, T. C., Brennan, A. M., Osofsky, H. J., Osofsky, J. D., and Lawrason, B. (2010). "Children of Katrina: Lessons Learned About Post-disaster Symptoms and Recovery Patterns." *Child Development*, 81(4), 1241–1259.
- Markhvida, M., Walsh, B., Hallegatte, S., and Baker, J. (2020). "Quantification of disaster impacts through household well-being losses." *Nature Sustainability*, 3(7), 538–547.
- Matyas, D., and Pelling, M. (2015). "Positioning resilience for 2015: the role of resistance, incremental adjustment and transformation in disaster risk management policy." *Disasters*, 39(s1), s1–s18.
- Nastasi, B. K., Jayasena, A., Summerville, M., and Borja, A. P. (2011). "Facilitating long-term recovery from natural disasters: Psychosocial programming for tsunami-affected schools of Sri Lanka." *School Psychology International*, 32(5), 512–532.
- Opabola, E. A., and Galasso, C. (n.d.). "A Probabilistic Framework for Post-Disaster Recovery Modeling of Buildings and Utility Networks." *Earthquake Spectra*.
- Opabola, E. A., Galasso, C., Rossetto, T., Meilianda, E., Idris, Y., and Nurdin, S. (2023). "Investing in disaster preparedness and effective recovery in education systems." *International Journal of Disaster Risk Reduction*, (in press)
- Platt, S., Gautam, D., and Rupakhety, R. (2020). "Speed and quality of recovery after the Gorkha Earthquake 2015 Nepal." *International Journal of Disaster Risk Reduction*, Elsevier Ltd, 50(June), 101689.

- Ratnayake, R., and Rameezdeen, R. (2008). "Post disaster housing reconstruction: Comparative study of donor driven vs. owner driven approach." *Women's career advancement and training & development in the*, Date 2008 USIR is a digital collection of the research output of the ..., 1067.
- Sadique, M. Z., Adams, E. J., and Edmunds, W. J. (2008). "Estimating the costs of school closure for mitigating an influenza pandemic." *BMC public health*, Springer, 8(1), 1–7.
- Sander, B., Nizam, A., Garrison Jr, L. P., Postma, M. J., Halloran, M. E., and Longini Jr, I. M. (2009). "Economic evaluation of influenza pandemic mitigation strategies in the United States using a stochastic microsimulation transmission model." *Value in Health*, Elsevier, 12(2), 226–233.
- Sanderson, D., and Ramalingam, B. (2015). "Nepal Earthquake Response: Lessons for operational agencies." *Alnap*, 30.
- Unnikrishnan, V. U., and van de Lindt, J. W. (2016). "Probabilistic framework for performance assessment of electrical power networks to tornadoes." *Sustainable and Resilient Infrastructure*, Taylor & Francis, 1(3–4), 137–152.
- Westoby, L., Wilkinson, S., and Dunn, S. (2021). "The road to recovery: Understanding the challenges affecting school reconstruction in rural Nepal following the 2015 Gorkha earthquake." *International Journal of Disaster Risk Reduction*, Elsevier Ltd, 56(February), 102120.

REVIEWER COMMENTS

Reviewer #1 (Remarks to the Author):

The authors have addressed all my comments, and the contributions and limitations of the paper seem to be better described. I would only suggest that the authors expand on point 2 regarding the non-physical dimension of recovery so that readers can better understand the strengths and limitations of risk models. Physical recovery is often linked to non-physical processes too, e.g., the decision to repair a building is human, after all. It would be good to expand on the psychosocial recovery and management structure, perhaps with examples from previous earthquakes. Otherwise, it is clear what the limitations are and how they can be addressed with future research. Additionally, the authors mentioned that recovery times and amplifications are based on limited data. But where can this data even be collected? Can data collected in other regions be directly or indirectly applied to a risk model? How? Addressing these questions will help the authors get their models used in other locations.

Reviewer #3 (Remarks to the Author):

Review for NCOMMS-22-42189A "Informing disaster-risk management policies for education infrastructure using scenario-based recovery analyses"

This manuscript presents a recovery modeling framework for education infrastructure (school facilities) to prioritize education continuity after future disasters. The authors present this framework using data collected on 80 school facilities in Central Sulawesi, Indonesia and showcase the capabilities of this framework for decision-making through several case studies. Importantly, they blend the modeling framework with stakeholder engagement to ground the proposed framework in reality.

While the reviewer appreciates the general approach for this work, in order to recommend this manuscript for publication in Nature Communications several areas need to be improved so the manuscript is at the caliber of this journal. Content-wise, more detail is needed to connect this framework to other literature and to support the statements that this is a novel framework that is useful and valid for decision-making. In addition, more background is needed on several methodological approaches and assumptions. Finally, overall clarity can be improved in the writing and in the figures.

Below I organize my comments in to Major and Minor. I underline sentences with suggestions that should be addressed.

Major Revisions

1. Clarify novelty of the recovery modeling framework: As mentioned by the previous Reviewer #1, I still do not fully understand whether the overall recovery framework presented in this article is novel. As it reads, the novelty of the framework seems to come in through the definition of specific characteristics within each module to be specific to educational facilities, but perhaps not additions to the overall recovery modeling framework. Does the novelty of this work exist in (a) additions to the overall formulation of the overall recovery modeling framework (i.e., adding new modules) OR is it solely (b) the modification of modules that have already been defined for other buildings to the educational sector? Please make explicit in the text with relation to previous studies that employ the recovery modeling framework.

Based on the authors' previous responses to Reviewer #1, part of this paper's contribution is (b), though additional justification is needed for those modified modules (see comments below).

2. Justify definition of the functionality indicator for school facilities: One of the main contributions of this manuscript is the description of post-disaster functionality for school facilities and the authors assert that education continuity is equal to physical continuity of the school buildings,

specifically stating that “continued access to education” is synonymous with “ratio of students with safe and occupiable classrooms” (lines 163-164). Because of the importance of this functionality definition to this work, the authors should justify that ratio of students with safe classrooms is an ideal indicator for education continuity, ideally referencing other literature or the authors’ referenced experience studying the schools in Palu. One can imagine multiple counterfactuals to this assertion: a scenario where education is not disrupted in wealthier households that can afford tutoring or a scenario where classes are held in alternative sites or without structures at all.

a. Additionally, the authors added in their revisions that “building functionality strongly depends on specific occupancy type” (line 84). However, the classification of functionality level (FL) from damage states (DS) would be the same for structures of all occupancy types, no? This may be a terminology issue (see minor comments below) or this statement needs additional explanation with referencing to other building functionality literature that asserts this (see comment #7 below).

3. Improve description of decision-making analysis and socio-political factors: according to the authors, the novelty of this framework is the ability to consider a multicriteria decision-making approach and the ability to consider sociocultural, political, economic, environmental, and technical factors that influence recovery. While the reviewer appreciates these contributions, as they are limitations in a traditional risk modeling framework, several areas need to be addressed to better understand this contribution.

a. Decision-making analysis: Can the authors be more specific on who they imagine would be users of the proposed model among the many decision-makers they interviewed? The reviewer can imagine that is more feasible for some organizations with more technical capacity to run and interpret these models (for example BNPB in Indonesia or INGO’s like the Red Cross) compared to community-based NGOs that are leading recovery.

b. Socio-political factors: Can the authors provide more description of how the time application factors were selected in this paper? In addition, it seems like these factors are more pessimistic than optimistic, are there any additional mitigation factors that could help promote school recovery?

4. Improve validation and discussion/interpretation of results: While the reviewer understands that validation can be difficult, it seems that the authors have enough experience and quantitative/qualitative data to point to the validity of the results to the experience of recovery in Indonesia (as evidenced by the final sentence on Line 485). This would help ensure that the modeling approach is valid. The reviewer would appreciate additional discussion on the validity of this model as reflected by observations on school recovery in Indonesia beyond this sentence. For example, why is School 15 ranked #1 and #2 in Scenarios 1 and 3 and 11 in Scenario 2? Does this make sense based on the characteristics of those schools?

5. Describe methods for focus group discussions and semi-structured interviews: Because a contribution of this paper is tying the model to realistic policies and applications that were defined through interviews with key stakeholders from schools, NGOs, the government, engineers, and contractors, more description is needed on these methods, recognizing that most readers can reference the other study published by the authors (ref. 36). Mainly, the reviewer would like to see the following in this article:

a. A brief discussion of the qualitative methods employed in the methods section of this paper, including the types of semi-structured interviews and the participant selection strategy, the number of participants in each sector mentioned, and the same for the focus groups (or workshops as mentioned in the referenced paper).

b. IRB reference number, and included in the editorial policy checklist.

6. Improve description for methodological decisions and assumptions: There were several modeling decisions and assumptions that were unclear in the text, and need to be addressed by the authors:

a. How were the 80 schools selected out of 2500 total schools? Why were these two districts selected?

b. Can the authors provide descriptive statistics of the 80 schools, specifically on the data needed for the recovery model? For example, according to Equation (1) you population per class and number of classes per building is required.

- c. Why are fragility curves from South Asia sufficient to use for Southeast Asia?
- d. How was the scenario selected, is it the same as the 2018 Central Sulawesi event? It seems like the authors compare results with values from the Sulawesi event (lines 354 – 362) so it would make sense to recreate that event.
- e. Any time an assumption is made, please justify, and please cite relevant literature if possible. For example, assumptions made on TLC's and closures on lines 527 – 529, is this from the semi-structured interviews? Or the decisions for priorities by age group of students in the school (lines 570-573). This happens multiple times and is difficult to track all.
- f. It is unclear what the differences are in minimal versus sufficient anticipatory budgets. Does the minimal budget distribute the 45% equally across schools? Please describe and justify decision.

7. Improve literature reviews and citations: The reviewer recognizes that there is a citation limit to Nature Communications articles, however, several citations are required for statements made in the manuscript.

- a. Lines 76 -77, studies highlighted that governments....
- b. Lines 80-82, please specify that these citations are on physical infrastructure recovery modeling. Also please acknowledge Alisjahbana et al 2022 reconstruction modeling (citation 33) here.
 - a. There is a robust literature on recovery trajectories broadly of educational programs that could be referenced, see Betty S Lai's work.
 - b. Again, citation for statement on functionality and occupancy on Lines 83-84.
 - c. Citation for "past events have shown that utility networks are quickly fixed following disasters" on line 148.
 - d. Semi-structured interviews in line 343.

Minor Revisions

1. Many of the figures are low quality, especially for this journal, many of them are blurry.
 - a. Figure 2 -- What are the red lines in the left, is that the outline of Central Sulawesi? Where is the NNW-SSE fault on this map?
 - i. Is it possible to show the distribution of shaking due to the selected scenario
 - b. Figure 3 – should be stated in the text that this is a hypothetical boundary
 - c. Figure 4 – "Reliance on foreign aid?"
 - d. Figure 5 – It is difficult to see the difference in line type between the two CDF's
 - e. Figure 6 – It would be easier to interpret if the x-axis was labeled directly with the time amplification factors.
 - f. Figure 7 – It would be easier to read if the weights were on the y-axis and horizontal.
 - g. Figure 9 – It would be easier to interpret the modeling framework if the module numbers described starting on line 154 were also labeled in Figure 9.
 - h. Table 1 -- It would be helpful to relate terminology used in Table 1 to that in Figure 1. For example, is reconstruction time the same as recovery time t_r ?
2. It would be helpful to be consistent with terminology:
 - a. The following terms seem to be used interchangeably, but it would be helpful to be as specific as possible here, so readers can understand what part of the model is being referred to: Functionality vs. recovery vs. resilience vs. students with access to...
 - i. Is it resilience or recovery modeling (line 105 vs line 113)
 - ii. When referring to a specific functionality indicator, please describe what the indicator is, especially in the methods, For example, in Equation (1) and Equation (2), what are the specific functionality indicators for Q_0 and q_0 . This will help readers understand functionality and its units.
 - b. Anticipatory budget versus anticipatory funds
3. Line 100, 103, abstract—Is the term "marginalized community" used synonymously with developing community? The whole of Indonesia is not marginalized, there are some very wealthy households and populations there. It only recently reverted back to lower-middle income. Would defer to "Global South" or "Lower-Middle Income" country instead.
4. Introduction – M7.0, M 7, or (M) 7.0? One should be used to be consistent
5. Line 247 -- Focused group discussions or focus group interviews?
6. The sentence from 330-332 seems like a random addition
7. The framing for sentence "One of the key recommendations..." in lines 377 – 378 is a little offputting, especially if someone in the Government were to read this.

8. Line 605 and 609 -- Table 4 should be Table 5?
9. Why is there an asterisk next to pandemic in Table 5?
10. The wording to describe Reference 34 in Line 607 makes it seem that the survey was done by other authors, not that the authors of this manuscript surveyed other authors' studies.

Response to comments for “Informing disaster-risk management policies for education infrastructure using scenario-based recovery analyses” by Eyitayo Opubola and Carmine Galasso

NCOMMS-22-42189A

We thank the handling editor and the two reviewers for their thoughtful and insightful comments, which have significantly improved the quality of the revised manuscript. The reviewer’s comments have been numbered and listed below (in *italic*), followed by our responses in **blue**. Extracts from the manuscript are reported in **green**, with additions to the revised manuscript in **bold**.

Reviewer #1 (Remarks to the Author):

The authors have addressed all my comments, and the contributions and limitations of the paper seem to be better described. I would only suggest that the authors expand on point 2 regarding the non-physical dimension of recovery so that readers can better understand the strengths and limitations of risk models. Physical recovery is often linked to non-physical processes too, e.g., the decision to repair a building is human, after all. It would be good to expand on the psychosocial recovery and management structure, perhaps with examples from previous earthquakes. Otherwise, it is clear what the limitations are and how they can be addressed with future research.

We want to thank the reviewer for the positive overall feedback on our revised manuscript. We appreciate the reviewer’s comments in terms of providing further remarks on the non-physical domain of school recovery and links between the physical and non-physical domains of recovery. We also agree that it is important to highlight the integration of physical and non-physical spheres of school recovery as a topic for future research.

The following modifications have been made to line 63:

There are two distinct domains of post-disaster school recovery necessary for education continuity – physical and non-physical. The former is related to the conditions of the physical infrastructure (e.g., classrooms; laboratories; water, sanitation, and hygiene (WASH) facilities; power and water utilities). The non-physical domain, for instance, is associated with the post-disaster management structure of a given facility (e.g., in terms of repair/replacement decision-making) and psychosocial recovery of school children and staff (e.g., Kronenberg et al., 2010; Nastasi et al., 2011). **Poorly-managed disaster-induced psychological disorders can influence changes in behavior, memory, and development of school children** (e.g., Kar, 2009), **thereby impacting education continuity. There are linkages between school recovery’s physical and non-physical (especially psychosocial) domains. As emphasized by past events, prolonged stay in temporary housing settlements and delayed recovery in school physical infrastructure can impact the long-term psychosocial well-being of school children** (e.g., Nastasi et al., 2011). We note that our study focuses on the analytical modeling of the physical infrastructure domain of school recovery. **Additional studies are needed to explore the efficient integration of physical and non-physical domains of school recovery in analytical recovery modeling frameworks.**

Additionally, the authors mentioned that recovery times and amplifications are based on limited data. But where can this data even be collected? Can data collected in other regions be directly or indirectly applied to a risk model? How? Addressing these questions will help the authors get their models used in other locations.

As discussed in Line 655, “Data on recovery times, mitigation and amplification factors are typically collected through post-disaster field observations, interviews and focus group discussions carried out by other authors who have compared pre- and post-disaster reconstruction projects in lower-middle income countries.” For example, Sharma et al. (2018) reported that the absence of local government resulted in several months of delays before post-disaster emergency support could reach out to specific regions affected by the 2015

Gorkha earthquake (Nepal). Furthermore, resolving fundamental issues (e.g., tender process, building permits, and land acquisition) took at least four times the required time. Data presented by Weerakoon et al. (2007) show that the recovery rate of buildings in Sri Lanka following the 2004 Indian Ocean tsunami was eight times larger in conflict zones compared to zones outside the conflict region. Reports (e.g., Gharaati and Davidson 2008; Kennedy et al. 2008) have highlighted that poor management skills can lead to construction delays, rejection by beneficiaries, rework, and demolition of newly constructed buildings. Such delays can impede the recovery time by a factor of up to three. In addition, Opabola et al. (2023) interviewed engineers, contractors, and NGO officials involved in the post-2018 Palu earthquake recovery. The interviews provided information on the increased time to complete various tasks relative to pre-disaster scenarios. All this information contributed to Table 5 in our paper.

Also, we recognize that recovery-impeding factors (and related data) vary for different events, countries, and local contexts. This is a big challenge for data transferability. To address this, the factors presented in the table are provided as ranges based on values from different countries of similar human development indices rather than single average or median values. It is intended that users can select a value within the range that best replicates their own local context and adopt it in the PERT model. The PERT model further allows users to simulate desired levels of pessimism in recovery time analyses. It is expected that risk modeling tools adopting this approach (including selecting relevant factors for their own scenario) will provide useful **insights** into **possible** future scenarios and their outcomes.

The following modifications have been made to line 655 to reflect these aspects:

Data on recovery times, mitigation, and amplification factors are typically collected through post-disaster field observations, interviews, and focus group discussions carried out by other authors who have compared pre- and post-disaster reconstruction projects in lower-middle income countries. **For example, Sharma et al. (2018) reported that the absence of local government resulted in several months of delays before post-disaster emergency support could reach specific regions affected by the 2015 Gorkha earthquake in Nepal. Furthermore, resolving fundamental issues (e.g., tender process, building permits, and land acquisition) took at least four times the required time. Data presented by Weerakoon et al. (2007) show that the recovery rate of buildings in Sri Lanka following the 2004 Indian Ocean tsunami was eight times larger in conflict zones compared to zones outside the conflict region. Reports (e.g., Gharaati and Davidson 2008; Kennedy et al. 2008) have highlighted that poor management skills can lead to construction delays, rejection by beneficiaries, rework, and demolition of newly constructed buildings. Such delays can impede the recovery time by a factor of up to three. Furthermore, Opabola et al. (2023) interviewed engineers, contractors, and NGO officials involved in the post-2018 Palu earthquake recovery. The interviews provided information on the increase in the time to complete various tasks relative to pre-disaster scenarios. All this information contributed to Table 5 in our paper.**

The following modifications have been made to line 686 to discuss data transferability issues:

Also, we recognize that recovery-impeding factors (and related data) vary for different events, countries, and local contexts. This is a big challenge for data transferability. To address this, the factors presented in the table are provided as ranges based on values from different countries of similar human development indices rather than single average or median values. It is intended that users can select a value within the range that best replicates their own local context and adopt it in the PERT model. The PERT model further allows users to simulate desired levels of pessimism in recovery time analyses. It is expected that risk modeling tools adopting this approach (including selecting relevant factors for their own scenario) will provide useful insights into possible future scenarios and their outcomes.

Reviewer #3

This manuscript presents a recovery modeling framework for education infrastructure (school facilities) to prioritize education continuity after future disasters. The authors present this framework using data collected on 80 school facilities in Central Sulawesi, Indonesia, and showcase the capabilities of this framework for decision-making through several case studies. Importantly, they blend the modeling framework with stakeholder engagement to ground the proposed framework in reality.

While the reviewer appreciates the general approach for this work, in order to recommend this manuscript for publication in Nature Communications several areas need to be improved so the manuscript is at the caliber of this journal. Content-wise, more detail is needed to connect this framework to other literature and to support the statements that this is a novel framework that is useful and valid for decision-making. In addition, more background is needed on several methodological approaches and assumptions. Finally, overall clarity can be improved in the writing and in the figures.

We would like to thank the reviewer for highlighting the value of our study. We appreciate the reviewer's comments in terms of further clarifying specific study assumptions and methodological approaches and more discussions on existing literature. We must note that it is our desire to expand/provide as many details as possible in the manuscript, but we are limited by the page limit requirements of the journal. However, we have made various changes to the manuscript content in response to the reviewer's comments. The comment responses and content modifications are described in depth below.

Clarify novelty of the recovery modeling framework: *As mentioned by the previous Reviewer #1, I still do not fully understand whether the overall recovery framework presented in this article is novel. As it reads, the novelty of the framework seems to come in through the definition of specific characteristics within each module to be specific to educational facilities, but perhaps not additions to the overall recovery modeling framework. Does the novelty of this work exist in (a) additions to the overall formulation of the overall recovery modeling framework (i.e., adding new modules) OR is it solely (b) the modification of modules that have already been defined for other buildings to the educational sector? Please make explicit in the text with relation to previous studies that employ the recovery modeling framework.*

Thanks for this very important comment. Indeed, our study builds on existing frameworks for other building types but tailors those existing frameworks to school facilities; in addition, we also include a novel multicriteria-decision-making (MCDM) module for inclusive school reconstruction prioritization, which to the best of our knowledge, is not included in any existing recovery modeling framework. Furthermore, while we acknowledge existing reconstruction-time optimization approaches (e.g., Alisjahbana et al. 2021), we believe an MCDM approach enables decision-makers to better account for many factors/criteria that impact recovery in marginalized communities and for their preferences towards those criteria. We have now provided more clarifications in Line 121 to reflect this.

We aim to fill the aforementioned gaps by (a) proposing a post-disaster recovery modeling approach for education systems; (b) demonstrating how the proposed approach can support stakeholders in quantifying the impact of policies (such as early response financing mechanisms and recovery management type) on education continuity. **The framework builds on existing community-level recovery frameworks developed for other infrastructure types** (Burton et al. 2016; Lin and Wang 2017; Opabola and Galasso 2023). However, the current study explicitly incorporates an approach to account for sociocultural, economic, political, technical, and environmental factors influencing the sustainable recovery of school physical infrastructure in low-income and lower-middle-income countries. **Firstly, the recovery time estimation module accounts for various recovery enhancing and impeding factors through a stochastic Program Evaluation and Review Technique (PERT), which enables users to simulate desired levels of pessimism/optimism on each task in**

the recovery process. Furthermore, the proposed approach embeds a novel multicriteria decision-making (MCDM) module for intervention prioritization. **In comparison with existing school reconstruction prioritization methods which just aim to minimize travel time for all affected students in the community** (e.g., Alisjahbana et al., 2021), **the proposed MCDM accounts for factors such as school level, land availability, site accessibility, and conditions of existing temporary learning centers, enabling decision-makers to better account for many factors/criteria that impact recovery in marginalized communities and for their preferences towards those criteria. The proposed approach can contribute to the community-level inclusive recovery of school physical infrastructure.**

***Justify definition of the functionality indicator for school facilities:** One of the main contributions of this manuscript is the description of post-disaster functionality for school facilities and the authors assert that education continuity is equal to physical continuity of the school buildings, specifically stating that “continued access to education” is synonymous with “ratio of students with safe and occupiable classrooms” (lines 163-164). Because of the importance of this functionality definition to this work, the authors should justify that ratio of students with safe classrooms is an ideal indicator for education continuity, ideally referencing other literature or the authors’ referenced experience studying the schools in Palu. One can imagine multiple counterfactuals to this assertion: a scenario where education is not disrupted in wealthier households that can afford tutoring or a scenario where classes are held in alternative sites or without structures at all.*

a. Additionally, the authors added in their revisions that “building functionality strongly depends on specific occupancy type” (line 84). However, the classification of functionality level (FL) from damage states (DS) would be the same for structures of all occupancy types, no? This may be a terminology issue (see minor comments below) or this statement needs additional explanation with referencing to other building functionality literature that asserts this (see comment #7 below).

Thanks for this very important comment. The adopted indicator aligns with Comprehensive School Safety (CSS) Targets and Indicators developed by the Global Alliance for Disaster Risk Reduction and Resilience in the Education Sector (GADRRRES). This has now been clarified. Regarding the scenarios mentioned by the reviewer, our adopted definition of classroom refers to both permanent and temporary learning facilities, which is reflected in our case studies. This study assumes that having a temporary learning center (TLC) (as contained in the CSS targets and indicators) is the minimum requirement for classroom quality. Regarding the scenario where education is not disrupted in wealthier households that can afford tutoring, this scenario is less relevant to marginalized communities, which is the main focus of this study.

The following modifications have been made to line 177:

The primary functionality indicator considered in this study is education continuity. **According to the Comprehensive School Safety (CSS) Targets and Indicators developed by the Global Alliance for Disaster Risk Reduction and Resilience in the Education Sector (GADRRRES) (GADRRRES 2015), post-disaster education continuity indicators include (a) duration of disaster-induced school closure; (b) the number of students displaced from schools; and (c) number of students in temporary learning centers. In line with the CSS indicators, we define functionality level as the proportion of students with continued access to education (either in a permanent or temporary learning center).** For each school, this is estimated as the ratio of students with safe and occupiable classrooms to the total number of students in the school.

We have also provided further clarification on what we mean by “building functionality strongly depends on specific occupancy type” at line 92:

We also note that post-disaster recovery modeling frameworks cannot be generic because building functionality strongly depends on the specific occupancy type. **For example, a moderately damaged**

residential building may be suitable as a shelter-in-place (meaning occupants are not entirely displaced). However, a similarly damaged building might be unsuitable for learning purposes.

Improve description of decision-making analysis and socio-political factors: according to the authors, the novelty of this framework is the ability to consider a multicriteria decision-making approach and the ability to consider sociocultural, political, economic, environmental, and technical factors that influence recovery. While the reviewer appreciates these contributions, as they are limitations in a traditional risk modeling framework, several areas need to be addressed to better understand this contribution.

a. Decision-making analysis: Can the authors be more specific on who they imagine would be users of the proposed model among the many decision-makers they interviewed? The reviewer can imagine that is more feasible for some organizations with more technical capacity to run and interpret these models (for example BNPB in Indonesia or INGO's like the Red Cross) compared to community-based NGOs that are leading recovery.

We thank the reviewer for this comment. The technical capacity needed to run the entire proposed recovery modeling framework is similar to that required for any risk assessment tool. As highlighted in Line 144, "The proposed approach is novel and adaptable to local conditions and different hazards. It can help education ministries and local disaster risk management authorities implement policies before a disaster to help maximize education continuity post-disaster." Specifically, the MCDM framework is straightforward. One of the reasons we chose MCDM over other mathematically intensive algorithms is the ease of adoption. MCDM analyses can be carried out, for instance, using Excel (e.g., see <https://www.youtube.com/watch?v=kfcN7MuYVeI&pp=ygUHdG9wc2lzlA%3D%3D>). We expect that local disaster risk management authorities, education ministries, and international NGOs will be able to use it easily.

The following modification has been made to Line 559 to reflect the reviewer's comment:

The intervention prioritization list was developed using the technique for order of preference by similarity to ideal solution (TOPSIS) (Hwang and Yoon 1981) – a multicriteria decision-making method (MCDM). **MCDM has been adopted in this study because of its simplicity. It can be easily implemented by relevant decision-makers (e.g., local disaster risk management authorities, education ministries, and international NGOs)** (Marcelo et al. 2016).

b. Socio-political factors: Can the authors provide more description of how the time amplification factors were selected in this paper? In addition, it seems like these factors are more pessimistic than optimistic, are there any additional mitigation factors that could help promote school recovery?

As discussed in Line 656, "Table 5 presents a range of recovery time mitigation and amplification factors based on a survey (Opabola and Galasso 2023) of observations, interviews, and focus group discussions carried out by other authors who have compared pre- and post-disaster reconstruction projects in lower-middle income countries." Furthermore, Figure 6 caption provides all the information on the adopted amplification factors for the analyses "Figure 6 – Median post-disaster reconstruction time for school buildings under (1) hostile political conditions; (2) pandemic; (3) land dispute; (4) poor management skills of contractors; (5) a combination of land dispute and poor management skills of contractors. The time amplification factors used in simulating the influence of these conditions in Eqs (3)-(5) are based on Table 5 in Methods. For each of the five scenarios, the optimistic time (a) was estimated using a time mitigation factor of 1.0, while the pessimistic time (b) is based on the upper-bound values of the time amplification factors in Table 5."

It is also noted that the adopted approach allows users to specify the PERT distribution based on their desired level of pessimism. This is discussed in line 677 “The most likely duration (m) captures the highest likelihood of completing a recovery phase in a given timeframe. If there is a higher likelihood that the mitigation factors would be more prevalent than the amplification factor, m is defined to be closer to a . Otherwise, m is defined closer to b . When uncertain, m can be defined as $0.5(a+b)$.”

Improve validation and discussion/interpretation of results: While the reviewer understands that validation can be difficult, it seems that the authors have enough experience and quantitative/qualitative data to point to the validity of the results to the experience of recovery in Indonesia (as evidenced by the final sentence on Line 485). This would help ensure that the modeling approach is valid. The reviewer would appreciate additional discussion on the validity of this model as reflected by observations on school recovery in Indonesia beyond this sentence. For example, why is School 15 ranked #1 and #2 in Scenarios 1 and 3 and 11 in Scenario 2? Does this make sense based on the characteristics of those schools?

We note that the probabilistic recovery time plot presented in Figure 5 was validated with field data, and this is described in the text. “We note that we interviewed 16 school principals with reconstruction projects in their schools and discovered that the completion time ranged from 180 – 400 days (Opabola et al. 2023). This

range is effectively captured in

Figure , providing some evidence of the efficiency of our proposed framework. On the other hand, as shown

in

Figure , agency-managed projects are likely to be completed two to four years after the disaster. We also assessed ongoing agency-managed reconstruction projects in Palu by interviewing NGO officials and

contractors. For an unnamed NGO with a significant number of school reconstruction projects in the Palu region, most reconstruction sites have not yet been reached four years after the disaster. More details are provided in Opabola et al. (2023). Hence, it may take up to five years or more to construct permanent structures in those schools.”

For clarity purposes, the discussions in the text have now been added to Figure 5

Figure 5 – Simulated cumulative distribution function (CDF) for the reconstruction time of community-managed and agency-managed school building reconstruction in the testbed community developed using the recovery time model presented in Equations (3) – (5). The CDF accounts for recovery-impeding (i.e., time amplification) factors that may influence both community- and agency-managed projects. The community-managed school building projects have a lower median recovery time because of little to no internal bureaucratic issues, more coordinated arrangements with relevant stakeholders, and little to no delay in a contract bidding process in these types of projects. **The range of completion times for community-managed building reconstruction in 16 schools in Palu (after the 2018 earthquake) is shown in the blue box. The green and red lines show the start and contract termination date for one of the prominent NGOs handling reconstruction projects in over 20 schools in Palu (see Opabola et al. 2023). (AMP – agency-managed project)**

Regarding the comment on the school prioritization list presented in Figure 8, we appreciate that the reviewer picked up on the changes in the ranking of School 15. We note that the combination of criteria weights for each scenario is intended to reflect how different groups of decision-makers may think. Scenarios 1 and 2 are cases where decision-makers prefer to prioritize schools that rely on temporary structures and schools with lower relative reconstruction costs (i.e., the cost of school reconstruction relative to the available budget), respectively. Scenario 3 is a scenario with no preference for any criterion. As shown in Figure 8a, School 15 is an elementary school that relies heavily on temporary structures, but the reconstruction project would take 27% of the entire budget. For scenario 1 (bias towards school reliance on temporary structures), school 15 gets prioritized. However, in scenario 2 - where the decision-makers choose not to prioritize the most capital-intensive projects - School 15 drops in the prioritization order. The MCDM allows decision-makers to visualize whether their bias (reflected in criteria weights) results in inclusive recovery.

The following modifications have now been added to Line 454 for clarification purposes.

Figure 7 looks at a case study where decision-makers are interested in looking at three scenarios. The first scenario is where decision-makers consider the level of a school’s reliance on TLCs as the primary criterion

for identifying schools that need to be prioritized for permanent structures (re)construction. The second scenario examines a case where affordability is considered the most important criterion, while the third scenario looks at a case where all four criteria are equally weighted. Figure 8a looks at defined performance metrics for the 15 damaged schools with average post-disaster functionality lower than 0.33 (i.e., the schools marked red in Figure 3). Further discussions on how the performance metrics are defined are provided in Methods. Using the proposed approach, Figure 8b provides the reconstruction hierarchy for each scenario. As shown in Figure 8b, the selected weights significantly influence the list. **As shown in Figure 8a, School 15 is an elementary school that relies heavily on temporary structures, but the reconstruction project would take 27% of the available budget. For scenario 1 (preference for schools relying on temporary structures), School 15 gets prioritized (Figure 8b). However, in scenario 2 - where the decision-makers choose not to prioritize the most capital-intensive projects, School 15 drops in the prioritization order. The MCDM can enable decision-makers to visualize whether their bias (reflected in criteria weights) results in inclusive recovery.**

Describe methods for focus group discussions and semi-structured interviews: Because a contribution of this paper is tying the model to realistic policies and applications that were defined through interviews with key stakeholders from schools, NGOs, the government, engineers, and contractors, more description is needed on these methods, recognizing that most readers can reference the other study published by the authors (ref. 36). Mainly, the reviewer would like to see the following in this article:

a. A brief discussion of the qualitative methods employed in the methods section of this paper, including the types of semi-structured interviews and the participant selection strategy, the number of participants in each sector mentioned, and the same for the focus groups (or workshops as mentioned in the referenced paper).

b. IRB reference number, and included in the editorial policy checklist.

We thank the reviewers for their comment. Information related to the stakeholder engagement exercises (including details of the area of inquiry, stakeholder engagement type, guiding questions, and the number of stakeholders) has been published to ensure their accessibility (See Opabola et al., 2023). Table 1 below is taken from the Opabola et al. (2023) study. Unfortunately, we cannot provide all the information in this paper because of page limitations and the fact that we want to avoid reproducing data already published in our previous articles.

Table 1: Adopted methods, questions, and respondents for stakeholders' engagement in Palu (adapted from Opabola et al., 2023)

s/no	Area of inquiry	Adopted method(s)	Guiding questions	Remarks/Stakeholders
1	Pre-2018 disaster preparedness level and post-2018 earthquake disruption to education continuity and recovery trajectory of schools	Semi-structured interview	Opabola et al., (2022)	Participants: 30 school principals
2	Successes and challenges of the school recovery projects	Semi-structured interview	Opabola et al. (2023)	Participants: Six NGO officials Two government officials Four civil engineers, and two building contractors
3	Identifying solutions to the preparedness challenges in the Central Sulawesi region	Focus group discussions	Opabola et al. (2023)	Participants: 30 school principals

4	Identifying solutions to the recovery challenges in the Central Sulawesi region	Focus group discussions	Opabola et al. (2023)	Participants: 36 participants four NGO officials, four government officials, eight civil engineers and building contractors, five school principals, and 15 university academics
---	---	-------------------------	-----------------------	---

We have added the following statement in Line 285 to point readers to the right information source.

Information related to the stakeholder engagement exercises (including details of the area of inquiry, stakeholder engagement type, guiding questions, and the number of stakeholders) are available online (See Opabola et al., 2023).

Also the UCL Ethics Project ID Number: ID280898 has been added to the Editorial policy checklist.

Improve description for methodological decisions and assumptions: *There were several modeling decisions and assumptions that were unclear in the text, and need to be addressed by the authors:*

a. How were the 80 schools selected out of 2500 total schools? Why were these two districts selected?

The following sentence has been added to Line 222 to address this comment:

The attributes of the schools (including population, location, and building characteristics) are heavily based on the extensive database of 2,500 schools collected by the authors in Central Sulawesi (Opabola et al. 2022a). The selected 80 schools represent the number of schools within two districts. The decision to select a relatively small testbed is based on the concept that disaster management decisions are generally made at the local government level (Melo Zurita et al. 2015). Furthermore, the small testbed enables the visualization of the impact of each considered policy at the building level rather than considering low-resolution information over a large grid area. **We note that the Central Sulawesi region is prone to cascading hazards. Given the focus of the case study on earthquake-induced ground shaking only, the main testbed selection criteria are the low potential for earthquake-induced cascading hazards (e.g., liquefaction, landslides, and tsunamis) and the reliability of available information on the schools based on data collected by the authors (Opabola et al. 2022a).**

b. Can the authors provide descriptive statistics of the 80 schools, specifically on the data needed for the recovery model? For example, according to Equation (1) you population per class and number of classes per building is required.

The following information has been added to line 240:

The number of buildings in the 80 schools ranges from one to nine, with a median of three and a standard deviation of 1.4. The average pupil population in each school building is 61, with a standard deviation of 16.5.

c. Why are fragility curves from South Asia sufficient to use for Southeast Asia?

Thanks for the comment. One of the challenges we faced was the lack of seismic fragility models for Palu schools. The absence of ground-motion data across the region following the 2018 earthquake made it impossible to also derive empirical fragility relationships. As highlighted in Line 256, "Due to the absence of fragility models for schools in Palu, we selected fragility models based on a review of published functions from similar archetypes in South Asia (Gautam et al. 2021; Giordano et al. 2021; Martins and Silva 2021).

Based on expert judgment, we concluded that the fragility models from these countries could be adopted in Palu. Additional studies are, however, needed to develop fragility functions for school buildings in Central Sulawesi.

The following sentence has been added to Line 256

Due to the absence of fragility models for schools in Palu, we selected fragility models based on a review of published functions from similar archetypes in South Asia (Gautam et al. 2021; Giordano et al. 2021; Martins and Silva 2021). **Based on expert judgment, we concluded that the fragility models from these countries could be adopted in Palu. In addition, we note that the study aims to showcase the proposed framework using realistic input data and discuss the relative effect of various disaster-risk management policies rather than performing a detailed/realistic risk assessment. Additional studies are, however, recommended to develop fragility models for school buildings in Central Sulawesi.**

D. How was the scenario selected, is it the same as the 2018 Central Sulawesi event? It seems like the authors compare results with values from the Sulawesi event (lines 354 – 362) so it would make sense to recreate that event.

Thanks for the comment. We believe that it is unlikely that we can adequately “recreate” the 2018 event because of (a) the lack of record ground-motion data for the case-study region and this specific event to validate simulated site demands - this is, in turn, reflected in the reliability of relevant shakemaps available for this event; (b) poorly constrained existing ground-motion models and site characterizations (i.e., soil types) for the region. We only compare the observed recovery time of collapsed buildings in Palu with the simulated recovery time. This is possible because the recovery time models are conditioned on the post-earthquake damage state, not the earthquake scenario (i.e., ground motion intensities). The following sentence has been added to Line 402 to reflect this aspect:

We can compare the observed recovery time of collapsed buildings in Palu with the simulated recovery time because the recovery time models are conditioned on the post-earthquake damage state and not the earthquake scenario (i.e., ground-motion intensities).

e. Any time an assumption is made, please justify, and please cite relevant literature if possible. For example, assumptions made on TLC's and closures on lines 527 – 529, is this from the semi-structured interviews? Or the decisions for priorities by age group of students in the school (lines 570-573). This happens multiple times and is difficult to track all.

The assumption is based on observations in Palu. Some of these observations are documented in a recent Earthquake Engineering Field Investigation Team (EEFIT) recovery mission to Palu by the first author (EEFIT 2022).

Line 586 has been modified to:

We assumed that all FL3 would initially be replaced by TLCs, followed by the construction of permanent structures (e.g., EEFIT 2022).

f. It is unclear what the differences are in minimal versus sufficient anticipatory budgets. Does the minimal budget distribute the 45% equally across schools? Please describe and justify decision.

The following clarifications have been added to Line 315:

Engagement with the school principals (Opabola et al. 2023) showed that some schools were able to carry out immediate post-disaster intervention using anticipatory budget set aside from the school operational assistance funding (locally referred to as Bantuan Operasional Sekolah - BOS) provided by the government.

More information on BOS can be found online (Rahmawati 2022). We observed that not all schools had this anticipatory budget, and this impacted their recovery.

Error! Reference source not found. illustrates two cases of available anticipatory budgets for the schools subjected to the considered M 7 scenario. **The first is a case where 45% of the 80 schools have an anticipatory budget. Without any explicit model to simulate the capability of principals to set aside an anticipatory budget, we randomly assigned an anticipatory budget to 36 schools in the testbed. In reality, such data can be collected by surveying school principals in a region of interest.** As shown in **Error! Reference source not found.a**, in a case where only 45% of the 80 schools have early response financing mechanisms, only a small fraction of the schools can ensure education continuity within two months, and the remaining schools may need to rely on foreign aid to construct TLCs, resulting in an undesirable recovery rapidity. On the other hand, a sufficient anticipatory budget (**i.e., all 80 schools have an anticipatory budget**) allows all the damaged school buildings to be immediately replaced by TLCs. For the case-study community, we show that sufficient anticipatory funding can reduce the recovery time for education continuity (through rapid construction of TLCs) by a factor of up to three.

Improve literature reviews and citations: *The reviewer recognizes that there is a citation limit to Nature Communications articles, however, several citations are required for statements made in the manuscript.*

a. Lines 76 -77, studies highlighted that governments...

Please note that our referencing style entails providing a piece of information in the first sentence before using subsequent sentences to provide evidence/conclusions from literature to back this up. In Line 84, for example, we state that **“However, various studies have highlighted that governments do not generally prioritize post-disaster education continuity, leading to severe education disruption or even termination. For example, many school children dropped out due to unavailable school infrastructure following the 2018 Central Sulawesi earthquake (e.g., Mahful et al., (2020)).”** The second sentence is used to provide citations that back the first sentence.

b. Lines 80-82, please specify that these citations are on physical infrastructure recovery modeling. Also please acknowledge Alisjahbana et al 2022 reconstruction modeling (citation 33) here.

Line 88 focuses on recovery modeling frameworks on infrastructure types other than education systems. Discussions on recovery modeling frameworks for education systems is presented in Line 106-113. The work of Alisjahbana et al. 2022 is indeed already referenced here: **“Fewer studies have sought to contribute to post-disaster recovery modeling in lower-middle income countries. For example, Alisjahbana et al. (2021) developed an optimization approach for school reconstruction scheduling by minimizing the sum of the distance all students in the region have to travel until all schools in the region are reconstructed.”**

a. There is a robust literature on recovery trajectories broadly of educational programs that could be referenced, see Betty S Lai’s work

Due to the journal’s editorial policies, we are unable to cite every work on the recovery trajectory of education systems.

With regard to the recommended references, our paper contains similarly relevant references (cited also by Betty S Lai’s work!) on psychosocial recovery in schools. However, we have now cited Lai et al. (2016). For example, in Line 63: “There are two distinct domains of post-disaster school recovery necessary for education continuity – physical and non-physical. The physical domain of post-disaster recovery is related to the conditions of the physical infrastructure (e.g., classrooms, laboratories, water, sanitation, and hygiene (WASH) facilities, power, and water utilities). The non-physical domain, for instance, is associated with the

post-disaster management structure and psychosocial recovery of school children and staff (Kronenberg et al. 2010; Lai et al. 2016; Nastasi et al. 2011). Poorly-managed disaster-induced psychological disorders can influence changes in behavior, memory, and development of school children (Kar 2009); thereby impacting education continuity. There are linkages between the physical and non-physical (especially psychosocial) domains of school recovery. For example, prolonged stay in temporary housing settlements and delayed recovery in school physical infrastructure can impact the long-term psychosocial well-being of school children (Nastasi et al. 2011).”

b. Again, citation for statement on functionality and occupancy on Lines 83-84.

c. Citation for “past events have shown that utility networks are quickly fixed following disasters” on line 148.

A reference has been provided for this statement. Line 165 has been modified as:

The interdependence of education and utility lifelines (e.g., water and power networks) is not discussed here because (a) past events have shown that utility networks are quickly fixed following disasters (Kuwata and Takada 2010); (...)

d. Semi-structured interviews in line 343.

A reference has been provided for this statement. Line 165 has been modified as:

The adopted times are based on outputs from semi-structured interviews with NGOs, engineering firms, and contractors actively involved in reconstruction projects in the Palu region (e.g., **Opabola et al., 2023**).

Minor Revisions

1. Many of the figures are low quality, especially for this journal, many of them are blurry.

All figures have been updated to increase their quality. The PDF conversion by the editorial platform resulted in low-quality/blurry figures in the original manuscript. We apologize for this.

a. Figure 2 -- What are the red lines in the left, is that the outline of Central Sulawesi? Where is the NNW-SSE fault on this map?

The red outline is Palu’s boundary. This has now been discussed in the Figure 2 caption as:

Figure 2 – Testbed school community. The map of Central Sulawesi is shown on the left. **The boundaries of Palu city are highlighted in red.** The relative location of all 80 schools in the testbed is shown at the top right; the distribution of the number of stories, age level, and design era of school buildings is shown at the bottom left. Pre-code and post-code buildings were constructed before and after the SNI 1726:2012 (Badan Standarisasi Nasional, 2012) building code.

The NNW-SSE fault is the Palu-Koro fault. However, the authors prefer not to refer to this fault directly in the text (or in the figure) because we want to avoid a case where readers assume we are trying to recreate the 2018 event, as discussed in the comment above. However, we have shown the fault location in Figure 2.

i. Is it possible to show the distribution of shaking due to the selected scenario

We thank the reviewer for their comment. Unfortunately, our paper has exceeded the page limit and the limit on the number of figures to accommodate new references and discussions. We will not be able to add more figures.

b. Figure 3 – should be stated in the text that this is a hypothetical boundary

The following text has been added in Line 243:

We note that the boundary around the 80 schools in Figure is hypothetical (to guarantee the anonymity of each specific school).

c. Figure 4 – “Reliance on foreign aid?”

Thanks for noticing the typo. The notation in Figure 4 has been modified to “Reliance on foreign aid”,

d. Figure 5 – It is difficult to see the difference in line type between the two CDF’s

One of the line types has been changed,

e. Figure 6 – It would be easier to interpret if the x-axis was labeled directly with the time amplification factors.

We thank the reviewer for the comment. The authors believe referencing the socio-political conditions here conveys the message better than just using the time amplification factors as the label. However, the amplification factors are now explicitly provided in the caption.

f. Figure 7 – It would be easier to read if the weights were on the y-axis and horizontal.

The weights now appear on the y-axis and horizontal in Figure 7.

g. Figure 9 – It would be easier to interpret the modeling framework if the module numbers described starting on line 154 were also labeled in Figure 9.

h. Table 1 -- It would be helpful to relate terminology used in Table 1 to that in Figure 1. For example, is reconstruction time the same as recovery time t_r ?

This was already defined in Line 650: “The average recovery time is assumed to be the sum of the average time required to inspect damaged buildings (T_{insp}), for the bidding and construction mobilization (T_{mob}), and to restore functionality through selected intervention process (T_{int}) – the three considered phases.”

2. It would be helpful to be consistent with terminology:

a. The following terms seem to be used interchangeably, but it would be helpful to be as specific as possible here, so readers can understand what part of the model is being referred to: Functionality vs. recovery vs. resilience vs. students with access to...

i. Is it resilience or recovery modeling (line 105 vs line 113)

The work focuses on recovery modeling. We have modified the text to reflect this. For example, Line 106 has now been modified to:

Studies have proposed simulation-based probabilistic frameworks to simulate the post-disaster **recovery** of education systems.

Line 116 has been modified to:

Fewer studies have sought to contribute to post-disaster **recovery** modeling in low-income and lower-middle income countries.

ii. When referring to a specific functionality indicator, please describe what the indicator is, especially in the methods, For example, in Equation (1) and Equation (2), what are the specific functionality indicators for Q0 and q0. This will help readers understand functionality and its units.

Line 567 has been modified as:

The post-disaster functionality of a school s (i.e., the proportion of students with access to a classroom in the school) given a simulated ground motion IM level resulting from an earthquake scenario EQ is then assessed as:

$$q_{0,j}^s | IM_j^s, EQ = \frac{n_{st_FL0,1}^s | IM_j^s, EQ}{n_{st_tot}^s} \quad (1)$$

where $n_{st_FL0,1}^s$ is the population of students with access to FLO and FL1 buildings in school s ($s = 1, 2, \dots, 80$ for the case study), $n_{st_tot}^s$ is the total population of students in school s , and j is the number of IM realizations ($j = 1, 2, \dots, 1000$ for the case study).

Line 573 has been modified as:

The community-level functionality (i.e., the proportion of students with access to a classroom in the community) given a simulated ground motion IM level resulting from an earthquake scenario EQ ($Q_{0,j} | EQ$) can also be expressed in terms of proportionality as:

$$Q_{0,j} | EQ = \sum_{s=1}^{n_s} q_{0,j}^s | IM_j^s, EQ \quad (2)$$

where n_s is the number of schools in the community.

b. Anticipatory budget versus anticipatory funds

The term “Anticipatory budget” has been replaced with “anticipatory funding” throughout the paper.

3. Line 100, 103, abstract—Is the term “marginalized community” used synonymously with developing community? The whole of Indonesia is not marginalized, there are some very wealthy households and populations there. It only recently reverted back to lower-middle income. Would defer to “Global South” or “Lower-Middle Income” country instead.

Thank you for your comment. All references to developing countries have been replaced with lower-middle-income countries (LMIC). As the reviewer rightly points out, there are wealthy communities in LMIC. The term marginalized community refers to communities confined to the lower end of the socio-economic spectrum in an LMIC, and not the entire country.

4. Introduction – M7.0, M 7, or (M) 7.0? One should be used to be consistent

We have now adopted M7.0 throughout the paper.

5. Line 247 -- Focused group discussions or focus group interviews?

The term ‘focus group discussion’ (FGD for short) is appropriate here.

6. The sentence from 330-332 seems like a random addition

The sentence was previously separated from its original position while responding to a reviewer’s comment. Line 369 now reads as follows:

“Typically, agency-managed projects are subjected to delays resulting from internal bureaucracy, agreements with the local authorities, community, and school authorities, and the lengthy bidding process for engineers and contractors. **Nevertheless, regardless of the management type, a given time is spent on damage assessment, clearing the site of rubbles (or identifying a relocation site), getting the relevant building permits, and so on.**”

7. The framing for sentence “One of the key recommendations...” in lines 377 – 378 is a little offputting, especially if someone in the government were to read this.

The sentence has been modified to

“**A key recommendation** from this case study is that government needs to understand that agency-managed projects may be delayed.”

8. Line 605 and 609 -- Table 4 should be Table 5?

We thank the reviewer for noticing the typo. The typo has now been fixed. Table 4 has been replaced with Table 5 in Line 663.

9. Why is there an asterisk next to pandemic in Table 5?

Typo. This has been removed.

10. The wording to describe Reference 34 in Line 607 makes it seem that the survey was done by other authors, not that the authors of this manuscript surveyed other authors’ studies.

The sentence has been modified to:

Error! Reference source not found. presents a range of recovery time mitigation and amplification factors based on a survey (Opabola and Galasso 2023) of observations, interviews, and focus group discussions **from published studies that have** compared pre- and post-disaster reconstruction projects in lower-middle-income countries.

References

- Alisjahbana, I., Graur, A., Lo, I., and Kiremidjian, A. (2021). “Optimizing strategies for post-disaster reconstruction of school systems.” *Reliability Engineering & System Safety*, Elsevier Ltd, 219(October 2021), 108253.
- BSN (Badan Standarisasi Nasional). (2012). *SNI 1726:2012, Tata cara perencanaan ketahanan gempa untuk struktur bangunan gedung dan non gedung*. Jakarta, Indonesia.
- Burton, H. V., Deierlein, G., Lallemand, D., and Lin, T. (2016). “Framework for Incorporating Probabilistic Building Performance in the Assessment of Community Seismic Resilience.” *Journal of Structural Engineering*, 142(8), 1–11.
- EEFIT. (2022). “Palu Day 3: School of Hard Knocks.” *Earthquake Engineering Field Investigation Team Recovery mission to Palu*, <<https://eefit.wordpress.com/2022/12/08/palu-day-3-school-of-hard->

- knocks/> (Jan. 5, 2023).
- GADRRRES. (2015). *CSS Targets and indicators and concept note for phase two*.
- Gautam, D., Adhikari, R., and Rupakhety, R. (2021). "Seismic fragility of structural and non-structural elements of Nepali RC buildings." *Engineering Structures*, 232, 111879.
- Gharaati, M., and Davidson, C. (2008). "Who knows best? An overview of reconstruction after the earthquake in Bam, Iran." *Proceedings of the*.
- Giordano, N., De Luca, F., Sextos, A., Ramirez Cortes, F., Fonseca Ferreira, C., and Wu, J. (2021). "Empirical seismic fragility models for Nepalese school buildings." *Natural Hazards*, 105(1), 339–362.
- Hwang, C.-L., and Yoon, K. (1981). *Multiple Attribute Decision Making*. Lecture Notes in Economics and Mathematical Systems, Springer Berlin Heidelberg, Berlin, Heidelberg.
- Kar, N. (2009). "Psychological impact of disasters on children: review of assessment and interventions." *World journal of pediatrics*, Springer, 5, 5–11.
- Kennedy, J., Ashmore, J., Babister, E., and Kelman, I. (2008). "The meaning of 'build back better': evidence from post-tsunami Aceh and Sri Lanka." *Journal of contingencies and crisis management*, Wiley Online Library, 16(1), 24–36.
- Kronenberg, M. E., Hansel, T. C., Brennan, A. M., Osofsky, H. J., Osofsky, J. D., and Lawrason, B. (2010). "Children of Katrina: Lessons Learned About Post-disaster Symptoms and Recovery Patterns." *Child Development*, 81(4), 1241–1259.
- Kuwata, Y., and Takada, S. (2010). "Business restoration related to lifeline after Tsunami disaster." *Journal of Earthquake and Tsunami*, 4(2), 73–81.
- Lai, B. S., Esnard, A.-M., Lowe, S. R., and Peek, L. (2016). "Schools and disasters: Safety and mental health assessment and interventions for children." *Current psychiatry reports*, Springer, 18, 1–9.
- Lin, P., and Wang, N. (2017). "Stochastic post-disaster functionality recovery of community building portfolios II: Application." *Structural Safety*, 69, 106–117.
- Mahful, R., Algifari, A., Jacobus, J., and Haqiq, A. (2020). "EPIC (Education Priority Compact) Strategy as A Solution for Education of Post-Disaster Refugee Children in Palu City, Central Sulawesi." *IOP Conference Series: Materials Science and Engineering*, 875(1), 012086.
- Marcelo, D., Mandri-Perrott, C., House, S., and Schwartz, J. (2016). "Prioritizing infrastructure investment: a framework for government decision making." *World Bank Policy Research Working Paper*, (7674).
- Martins, L., and Silva, V. (2021). "Development of a fragility and vulnerability model for global seismic risk analyses." *Bulletin of Earthquake Engineering*, 19(15), 6719–6745.
- Melo Zurita, M. de L., Cook, B., Harms, L., and March, A. (2015). "Towards new disaster governance: Subsidiarity as a critical tool." *Environmental Policy and Governance*, Wiley Online Library, 25(6), 386–398.
- Nastasi, B. K., Jayasena, A., Summerville, M., and Borja, A. P. (2011). "Facilitating long-term recovery from natural disasters: Psychosocial programming for tsunami-affected schools of Sri Lanka." *School Psychology International*, 32(5), 512–532.
- Opabola, E. A., and Galasso, C. (2023). "A Probabilistic Framework for Post-Disaster Recovery Modeling of

Buildings and Electric Power Networks in Developing Countries.” *Reliability Engineering & System Safety* (under review).

Opabola, E. A., Galasso, C., Rossetto, T., Meilianda, E., Idris, Y., and Nurdin, S. (2023). “Investing in disaster preparedness and effective recovery in school physical infrastructures.” *International Journal of Disaster Risk Reduction*, 103623.

Opabola, E. A., Galasso, C., Rossetto, T., Nurdin, S., Idris, Y., Aljawhari, K., and Rusydy, I. (2022a). “A Mixed-Mode Data Collection Approach for Building Inventory Development: Application to School Buildings in Central Sulawesi, Indonesia.” *Earthquake spectra*, ((In Press)).

Opabola, E., Aljawhari, K., Galasso, C., Rossetto, T., Nurdin, S., Meilianda, E., Idris, Y., and Rusydy, I. (2022b). *An inventory of school buildings in Central Sulawesi (Indonesia) developed through a mixed-mode data collection approach [Data set]*. Zenodo.

Rahmawati, E. (2022). “Definition of School Operational Assistance Fund and Amount Received.” <<https://nusamandiri.info/bantuan-operasional-sekolah/>> (Nov. 4, 2022).

Sharma, K., Apil, K. C., Subedi, M., and Pokharel, B. (2018). “Challenges for reconstruction after M w 7.8 gorkha earthquake: A study on a devastated area of Nepal.” *Geomatics, Natural Hazards and Risk*, Taylor & Francis, 9(1), 760–790.

Weerakoon, D., Jayasuriya, S., Arunatilake, N., and Steele, P. (2007). *Economic challenges of post-tsunami reconstruction in Sri Lanka*. ADBI Discussion Paper.

REVIEWERS' COMMENTS

Reviewer #1 (Remarks to the Author):

I thank the authors for addressing my comments. I recommend the paper for publication.

Reviewer #3 (Remarks to the Author):

Thank you to the authors for thoroughly considering and addressing my comments. This manuscript represents an impressive body of work on developing a recovery modeling framework for education systems, importantly considering an important context (LMIC) with an approach grounded in reality (through previous stakeholder engagement conducted in Indonesia). The contributions are clearer as well, in that they modify and add to previous community-level recovery frameworks to make them more applicable to education systems. I have no further comments.